# Licensed Professionals and Intergenerational Big-, Meso- and Micro-Class Immobility within the Upper Class; Social Closure and Gendered Outcomes among Italian Graduates

Lucia Ruggera * and Jani Erola

INVEST Sociology, Department of Social Research, University of Turku, 20014 Turku, Finland
* Correspondence: lucia.ruggera@utu.fi

**Abstract:** This article examines how processes of social closure promote persistence at the top of the occupational hierarchy and how they vary by gender. We focus on the links between professional closure strategies and intergenerational immobility in professional employment among Italian graduates. Italy displays the highest levels of service market regulation across Europe, and professionals are the largest occupational group within the upper class; therefore, it is crucial to analyse the link between professional closure and labour market outcomes among Italian graduates. Using ISTAT's survey on Italian graduates' labour outcomes and replicating the analyses of men in the ILFI survey, the origin-destination association is investigated at the big-, meso-, and micro-levels. We employ log-linear nested models and logistic regressions. The SPL sample offers a unique opportunity to analyse social mobility at the beginning of professionals' careers and provide in-depth explanations of the micro-level dynamics of social reproduction. The analyses indicate that children of regulated professionals have a higher propensity to follow in their parents' footsteps (micro-classes). Self-employment among professionals strongly increases intergenerational immobility at the top of the occupational hierarchy. The findings demonstrate that the combination of specific parental resources strongly helps professionals' sons and daughters to avoid social demotion.

**Keywords:** social closure; micro-class; Italian graduates; upper class; professional regulation; self-employment

## 1. Introduction

Previous literature has documented that intergenerational social immobility at the top of the social strata is high in many European countries, which is due to the disproportionate chances of professionals' children persisting within this stratum (Jonsson et al. 2009; Grusky 2005). In this article, we examine how social and professional closure affects the upper-class intergenerational immobility in Italy and how they vary by gender. Traditional social mobility studies have been criticized as being inherently descriptive and missing many explanatory, often gendered, mechanisms of socioeconomic inheritance (Breen and Goldthorpe 1997). To overcome these limitations, a suggested approach has been to focus on the so-called micro-classes (Weeden and Grusky 2005, 2012). While the benefit of this approach has sometimes been questioned (e.g., Goldthorpe 2002), it should be particularly beneficial when focusing on specific professions such as licensed professions[1]. First, advantages often associated more broadly with the upper social classes can be limited to professional occupations and larger entrepreneurs (e.g., Ruggera and Barone 2017). Second, the long-term trend of gender desegregation in labour markets seems to be largely explained by the influx of women into professional occupations where persisting labour market obstacles, which still hinder women's achievements, are considerably less obstructive than those in other occupational groups (Grusky and Marybeth 2001). Third, professional associations increase the competition over advantages with other occupational groups belonging to the same broader social class (Sørensen 2000). Thus, we stress

the importance of disaggregating the upper social class into more detailed categories to shed light on the crucial dimensions undermining its strong heterogeneity (Patterson et al. 2003). In order to test these assumptions, we use a sample of higher-education Italian graduates (ISTAT's survey on the labour market outcomes of Italian graduates 2011 (ISTAT 2011)—henceforth SPL 2011). Within the sample, we analyse the intergenerational social immobility of different fractions of the upper class with topological log-linear models. Indeed, the sample offers a unique opportunity to analyse social mobility at the beginning of professionals' careers. It provides for more in-depth explanations of the micro-level dynamics of social reproduction at the top of the occupational hierarchy. The SPL analyses (2011) are replicated with ILFI 1997–2005—a longitudinal survey on Italian families (a sample representative of the entire population) looking at first occupations—whose results are consistent with the SPL (2011). We employ big-, meso-, and micro-level approaches, which distinguish classes within the Service class in different ways. The standard big-class approach, traditionally used in many social mobility studies, only distinguishes higher and lower service classes (Service class I and II, see Erikson and Goldthorpe 1992). We are mainly interested in the former, which is distinguished from the latter by greater responsibilities in decision-making and greater rewards. Service class I can be further divided into three meso-classes: larger entrepreneurs, higher-level managers, and higher-grade professionals. There is rich evidence that professionals as a whole differ from managers and entrepreneurs in terms of value orientations, political attitudes, and lifestyles (e.g., Dalton and Klingemann 2007). Professionals associate more often with members of other professional groups (Lambert and Griffiths 2011). If cultural and social resources characterize the meso-class of professionals, managerial skills, greater autonomy and responsibilities, and higher rewards relate to managers and entrepreneurs (Erikson and Goldthorpe 1992); property distinguishes larger entrepreneurs from higher-grade managers. However, only the detailed distinctions between the micro-classes allow us to test our hypotheses on intergenerational social immobility and professional closure. A micro-class is defined as 'a grouping of technically similar jobs that is institutionalised in the labour market through different means, such as an association or union, licensing or certification requirements, or widely diffused understanding regarding efficient or otherwise preferred ways of organizing production and dividing labour' (Grusky 2005, p. 66). Micro-classes embody mechanisms (e.g., closure) and traits (e.g., culture) that emphasize occupation-specific resources transmitted from parents to children differently (Erikson and Goldthorpe 1992; Grusky 2005; Jonsson et al. 2009; Ruggera and Barone 2017).

We proceed with the analysis as follows. The upper class is disaggregated into three distinct mechanisms operating at different levels (big-, meso-, micro-) of social classes. We further enucleate self-employed professionals, which are mainly compared with *simply* employed professionals. Finally, gender variation is assessed at each analytical level, while the professional regulation and self-employment dimensions are evaluated at both the meso- and micro-levels.

## 2. Why Do Sons and Daughters of Professionals Feel Inclined to Become Professionals?

The children of professionals are more likely to assume professional than non-professional positions of equivalent social standing, which are defined as managerial positions within the service class (Weeden and Grusky 2012). According to Jonsson et al. (2009), the intergenerational transmission of occupation-specific economic, cognitive, cultural, and social capital fosters micro-class immobility in professional employment (see also Erola et al. 2022). Occupational specialised skills transmitted by families to their children are particularly relevant when competing for a specific profession. For example, the transfer of technical skills, such as drawing and building houses and bridges using Lego blocks, may be useful in technical professions like engineering and architecture. Thus, the children of parents working in such professions might have a greater advantage in specific subjects in school and be able to complete more courses on technical skills training. These skills are part

of a broader socialisation connected to the exposure of children to the occupational cultures of their professional parents, resulting in the transmission of profession-specific aspirations, values, and ethics. Parallel to on-the-job-skills, children of professionals may also develop specific personality characteristics and proactivity that are appreciated by employers in their corresponding fields. For example, the occupational culture of lawyers celebrates argumentation and rhetoric, which are characteristics the children of such professionals tend to appreciate (Ruggera and Barone 2017).

According to Jonsson et al. (2009), professional cultures are formed and maintained around two main conditions: a training regimen that transmits a set of values and way of life and some types of closure processes that ensure that professional members interact principally with one another. Similarly, lawyers undergo intensive training in law school and frequently interact with each other in a relatively closed workplace (i.e., the closure condition), thus creating and sustaining an occupational culture. Therefore, even if professionals' children are exposed to a *wider class culture* that likewise prepares them for professional destinations (i.e., abstract argumentation), it is possible to underlie the occupation-based transmission of cultural and cognitive resources when considering specific professions. Social capital matters because on-the-job social networks and the contacts of the parents provide children with information about employment opportunities and assistance with practical job-related matters over time (Jonsson et al. 2009). When considering economic resources, the inheritance of property within self-employment (also defined as fixed resources) is an exemplar factor that encourages social class immobility, as in the case of children of larger entrepreneurs (meso-class), reflecting the endeavour to avoid social demotion (Breen and Goldthorpe 1997). If following the rational choice perspective, the safest strategy for children of professionals to achieve that goal is to rely on the competitive advantages related to the occupation-specific resources of the family of origin (Jonsson et al. 2009).

These micro-dynamics of professional social reproduction—the transmission of cultural, social, cognitive, and economic resources from parents to children—can be studied by applying the micro-class approach to social mobility. This entails an analytical shift in focus from the overall amount of family resources available to children to more qualitative advantages. The micro-class approach assumes that rather than just larger scale classes, the labour market is divided into occupational groups with each one sharing specific sociocultural resources and social identities. These groups can and do act to further their collective economic interests (Sørensen 2000; Murphy 1988) by closing off opportunities to outsiders who belong to other social classes, creating conflict between the professional groups (Parkin 1979). The prediction does not specify gender differences in the closure processes (Larson 1977).

Sørensen (2000) argues that regulated professions create artificial monopolies in the service labour market by, for example, disregarding gender inequalities, which limits the level of internal competition. Consequently, this generates rents to insiders; advantages derived from rents concern all members of the profession. These processes involve professional associations, which operate as interest groups that direct their actions to policy makers. Associations of professionals impose restrictions based on credentials requirements (educational titles, training, and licenses), shrinking the pool of candidates, and code of conduct, legitimising artificial professional monopolies (Collins 1979).

On the one hand, credentials increase competitive struggles with other professional groups over market niches. On the other hand, restrictions promote equalitarian values within a micro-class, implying a replacement of social exclusion based on ascribed criteria (Abbott 1988). For example, in studies on earning inequality, those who can be defined as insiders—both men and women—benefit from collective action strengthening their economic rewards and social prestige (Bol and Weeden 2015; Weeden 2002). Thus, we should expect gender differences in mobility to be smaller among professionals, and that daughters of such professionals tend to reproduce the micro-class of their fathers as much as men do.

**H1.** *Sons and daughters of highly regulated professionals are more incentivised to follow in their parents' footsteps than those of other social classes.*

Closure strategies drive social reproduction in specific professions, fostering transmissions of family occupational resources towards the sons and daughters. Parents who are members of regulated professions tend to inculcate their values, ways of life, and cultures to their children because they can be potential new members of the same occupational groups. Indeed, we argue that both sons and daughters are socialised in the professional worlds of their parents from very early ages, lowering gender differences through specific cultural capital transmission (Bourdieu 1979; Murphy 1988).

It is well known that sociocultural gender differences exist, with class-specific socialisation patterns and cultural beliefs about gender leading to different expectations, educational pathways, and occupational careers for men and women (Ruggera 2016). Research applying the micro-*educational* stratification approach, which also covers the fields of studies that lead to the professions considered in this study, asserts that occupational community and social closure theories (at the micro-level) better explain the educational choices of men than those of women (cf. Grusky and Marybeth 2001). Women seem to be endowed with more class-wide generic skills through family socialisation, leading their educational pathways and occupational careers to be better explained at the big- and meso-level (see Ruggera and Barone 2017). Thus, micro-class and micro-education research suggest that these mechanisms could contribute to gender differences in opposite ways, enhancing them in education but weakening them in occupational attainment.

According to Erikson and Goldthorpe (1992), 'property' is a main characteristic of larger entrepreneurs, whereas professionals are defined by other means of closure, such as reliance on educational credentials and knowledge (meso-classes). Previous research on social mobility in Italy has shown that property inheritance tends to be more important for men than women (Schizzerotto 2002). Moreover, research at the international level assessed the low presence of women in the highest levels of corporate power (Grusky and Marybeth 2001).

However, inherited wealth may be particularly important for the social reproduction of some groups of professionals and for both men and women. Fixed economic resources, such as those of a law firm, are often accessible to the children of self-employed professionals, who also have good chances of inheriting the business along with its client portfolio (see Pellizzari and Orsini 2012). For regulated professions, this type of inherited wealth may be important as it may enhance the rewards that follow from the social closure. According to rational action theory, when families make attainment related decisions for their children, they tend to prioritise avoiding social demotion before upward mobility (Breen and Goldthorpe 1997). For the children of professionals, the safest strategy to achieve this is to rely on the competitive advantages related to the occupation-specific resources of the family of origin (Jonsson et al. 2009). If other professional occupations are less accessible for women than men, inherited business wealth may be important for status maintenance for both sexes among professionals. Thus, we expect that:

**H2.** *Social immobility is higher for self-employed regulated professionals than unregulated professionals.*

Micro-classes are not the sole mechanism of social immobility at the top of the occupational hierarchy (e.g., Ruggera and Barone 2017): Goldthorpe (2002) emphasised that not all children of doctors want to become a doctor; they might not develop aspirations and preferences for that specific profession. Accordingly, literature has demonstrated that professionals as a broader meso-class differ from managers and entrepreneurs in terms of value orientations, political attitudes, and lifestyles (e.g., Grusky 2005). Professionals associate more often with members of other professional groups (Larson 1977) who can also act as network resources for their children. The skills and cultural resources of the family also play a crucial role; even if children do not graduate in the same professional domain as their parents, parental background and cultural proximity may help them exploit

educational credentials to remain in the same meso-class rather than moving into other positions within the upper class (Ruggera and Barone 2017; Ruggera 2021).

There is little evidence on gender inequality in professionals' social reproduction in Italy (Chiesi 1997; Checchi 2010). This is true when maintaining a meso-class position is considered as part of a more general strategy of intergenerational immobility at the top, functioning as *safety nets* for those children who leave their micro-class of origin. It is plausible to expect that gender differences in immobility are higher on lower parts of the occupation ladder than among the upper class (specifically with the professional meso-class) because of two main reasons. First, regulated professionals are incubators of gender-egalitarian values; the broader meso-class of professionals should similarly act as safety nets for sons and daughters of professionals. Second, attaining tertiary education is the main compulsory entry barrier to licensed professions, and both men and women can exhaust profession-specific parental resources to gain access to them. Thus, we expect:

**H3.** *In professional employment, parental resources for those who leave their micro-class of origin increase persistence at the top, similarly boosting big-class rigidities for both men and women.*

### 3. Professional Closure in Italy

Regulation in professional employment increases rewards and advantages in the labour market, which in turn affects intergenerational mobility within the upper class. Thus, the relevance of micro-class structuration at the top reflects the strength of the relationship between education and the labour market and varies accordingly across countries (Ruggera and Barone 2017; Ruggera 2021). For example, German and Danish educational systems, even if to different degrees, emphasise occupational specificity. In addition, access to occupations is closely related to the possession of specific vocational certificates along the entire occupational ladder (Pellizzari and Orsini 2012). Moreover, these countries also have a well-established tradition of professional associations that have managed to impose and preserve strict access. At the opposite extreme, Sweden has a more comprehensive educational system in which vocational training is undeveloped and professional regulation is weak. Italy can be placed in between these two extremes. First, it has a comprehensive educational system in which vocational training is not as well developed as in Sweden. Second, it has strong professional associations that impose restrictions to accessing professions, such as educational credentials, which remind us of the occupational specificity of Germany or Denmark (Ruggera 2016, 2021). However, as shown in Table 1, the level of professional regulation in Italy is even higher than that in Germany. Hence, Italy is an exceptional case where professional regulation and related increasing rewards in the labour market strongly incentivise intergenerational immobility at the top.

**Table 1.** The index of entry-market regulations by country and profession (2013).

|  | Engineers | Architects | Legal Professions | Accountants | Pharmacists * | Doctors ** |
|---|---|---|---|---|---|---|
| Denmark | 0.0 | 0.4 | 3.7 | 1.9 | 2.3 | 3.5 |
| Germany | 2.7 | 4.1 | 5.2 | 5.8 | 1.6 | 3.7 |
| Sweden | 0.0 | 0.0 | 0.0 | 2.2 | 6.0 | 3.5 |
| Italy | 3.9 | 3.9 | 4.0 | 4.1 | 4.8 | 4.3 |

* it has calculated only in 2003. ** authors' own calculation.

The OECD's index is a useful way to summarise entry-market professional regulations, reflecting access restrictions and their relative importance. Professionals taken into consideration are engineers, architects[2], accountants, lawyers, pharmacists, and doctors. The index does not cover the so-called new professions, such as social workers or nurses. These occupations generally belong to the lower service class in the EGP social class scheme. Similarly, scientific and social science professions are not covered by this index because the level of regulation is virtually absent. OECD's index ranges from 0 to 6, where 6

reflects the maximum level of professional legal closure. This index is based on entry requirements accounting for 40% of the overall amount, including the university courses or other higher degrees needed to access the profession, duration of compulsory practice, number of professional exams, and number of entry routes to each profession. Licensing counts for another 40% of the index score, defined as the number of exclusive and shared tasks in each professional field, and the remaining 20% concerns the existence of quotas for each profession.

While entry restrictions increase the value of the occupation resources of the family of origin, excluding outsiders, they lower gender inequality, functioning as gender equality incubators for insiders (Weeden 2002). Thus, as theorised by closure theorists, exclusion based on educational credentials is further reinforced by other barriers leading to a licensing system (e.g., compulsory training and state exams).

## 4. Data, Variables, and Methods

For the main analyses, we used 'Italian graduates' labour market outcomes' survey collected by ISTAT (SPL 2011—see https://www.istat.it (accessed on 25 May 2016) for more information on different years surveys 'Sbocchi professionali dei laureati'). It is targeted towards graduates four years after graduation, thus providing a large sample of graduates from which we extrapolate 19.554 valid cases at the beginning of their professional careers, with detailed information on their education, family background, and self-employment. By definition, to properly study the propensity of licensed professionals' children to follow in their parents' footsteps, we do not consider those graduates who only hold a BA degree. However, because the data are, to some extent, biased towards upper-class offspring (Schizzerotto 2002), we extended the scope of our robustness analyses to include the full population using the Longitudinal Survey on Italian Families—surveys ILFI 1997–2005 providing a nationally representative sample containing 7167 valid cases for the origin-destination association. More information on the sample design can be found in Pisati and Schizzerotto (2004). The robustness of the results obtained with SPL (2011) is confirmed by considering professionals in the ILFI data 1997–2005—the log-linear analysis also in terms of micro-classes is conducted on men only (3.840 valid cases).

We used the EGP-class scheme[3] to analyse differences between social classes in both datasets. The class groupings are defined as follows: higher service class (I), lower service class (II), routine non-manual workers (IIIab), self-employed workers (IVab) and farmers (IVc), skilled manual workers (V-VI), and unskilled manual workers (VII). The three meso-classes of entrepreneurs (with at least nine employees), high-level managers (supervising at least nine employees), and high-level professionals comprise the higher service class (Erikson and Goldthorpe 1992). The micro-classes of professionals are classified as follows:

(a)　technical professions (architects and engineers);
(b)　professionals in life sciences (including pharmacists);
(c)　medical doctors;
(d)　legal professions (lawyers, judges, and notaries);
(e)　professionals in economics (including accountants[4]);
(f)　non-regulated professionals, including social scientists and professionals in scientific fields[5].

The SPL (2011) provides information on state license, professional training (also defined using a specific Italian term called *praticantato*) and specialisation in an employed position, all within specific occupational positions as a combination of CP2011[6], occupational codes and self-employment[7]. Hence, integrating all the professionals' information in the beginning of their professional career leaves very little room for micro-class misclassification. These micro-class distinctions match the same occupational categories of the OECD index of professional regulation, as reported in Table 1.

We fit a sequence of log-linear models that allowed us to control for the marginal distributions of origins and destinations to estimate relative immobility propensities. We

model immobility with topological log-linear models (Xie 1992), differentiating the diagonal cells of the mobility table at each of the 17 levels. Since our research questions focus on immobility within the higher service class, the parameters for the other parts of our table are fitted by controlling margins at the big-class level.

To test our hypotheses on immobility, we first fit models using a design matrix that distinguish the differences between immobilities, that is, separating estimates for the diagonal cells, at the big-class level. This and other design matrices for immobility are reported in the Appendix A. We then fit models further differentiating immobility within the Service class at the meso-level, thus differentiating immobility among entrepreneurs, managers, and professionals. Subsequently, we fit models that distinguish immobilities at each level, including micro-classes. Finally, to assess our hypotheses on the importance of regulation as a means of closure and self-employment, we fit an additional set of models where we first distinguished the regulated service class occupations, and then the self-employed at the meso- and micro-class levels. The nested structure was useful for analysing the professionals' social immobility because we assessed the heuristic value of the three analytical approaches and the effect of the self-employment dimension.

All models were fitted twice to assess gender differences. Immobility is defined as *homogeneous* when gender differences are held constant; whereas they are *heterogeneous* when they include gender interaction.

Finally, to complete the analysis related to the third hypothesis, logistic regressions are employed to estimate the chances of remaining in the higher service class by micro-, meso-, and big-class of origin. The propensity of persisting at the top of the occupational hierarchy was calculated by social class of origin conceptualised at different levels (e.g., meso and micro-classes), net of social and demographic control variables (age, secondary education grades, nationality, geographical area). Logistic regression models are employed because they allow considering a dichotomous outcome variable and different specifications of the explicative variable (social class of origin). The results can be easily interpreted in terms of probabilities due to the presentation of average marginal effects (AME).

## 5. Results: Gender Differences in the Micro-Class Social Reproduction at the Beginning of a Professional Career

Table 2 reports the fit indices of the above-described sequence of log-linear models using the SPL (2011) survey. For comparisons among the nested models (e.g., big-class structure net of meso- and micro-classes), we used likelihood ratio tests (column 4) that can be used to contrast models in terms of their fit (expressed by the deviance in column 1) and parsimony (degrees of freedom in column 2). Further, we report the dissimilarity index (column 3), that is, the percentage of cases misclassified by each model. We strongly rely on this measure for model selection, even if the attention is paid to all of these indices as a combination. Panel A in Table 2 considers differences in mobility at the big-, meso-, and micro-class levels, with and without assuming gender differences. Panel B reports the models considering our hypotheses on professional regulation and self-employment, at both the meso- and micro-class levels and once again considering gender differences.

The baseline model of conditional independence assumes—unrealistically—that there is no association between social origins and destinations. We used it only for the comparison of the fit indices for the rest of the models, specifically for the first two models regarding the standard big-class models to social mobility. Models la and 1b are yardstick models assuming standard immobility for the higher service class relevant for us to add further model distinctions at the meso- and micro-class levels. The design matrix fits each cell of upward mobility, downward mobility, and immobility at the big-class level; the cells belonging to the higher service class are collapsed. Thus, it assumes that immobility in the service class is the same across the meso- and micro-class groups within it. If Model 1a is compared to the fit statistics of the baseline model, one can see that the reduction of the dissimilarity index is substantial (from 0.94 to 0.58) because the model now captures all social (im)mobility at the big-class level. Unsurprisingly, the comparison between Models 1a and 1b

confirms gender differences at the big-class level: assuming gender differences in Model 1b further improves the dissimilarity index to 0.036. This is further confirmed with the chi-squared test reported in the fifth column ($L^2 = 1681.4 - 1546.6 = 134.8$; df $= 425 - 338 = 87$; $p = 0.000$).

**Table 2.** Fit indices of log-linear models of quasi perfect mobility with big-, meso- and micro-class rigidities with regulation and self-employment specification (N. 19.554).

| Model Description | $L^2$ | d.f. | Δ | Significance |
|---|---|---|---|---|
| Panel A | | | | |
| 0. Conditional independence | 2556.3 | 512 | 0.094 | - |
| *Big-class rigidities* | | | | |
| 1a. + big-classes (Homogeneous) | 1681.4 | 425 | 0.058 | 0.000 (M.0) |
| 1b. + big-classes (Heterogeneous) | 1546.6 | 338 | 0.036 | 0.000 (M.1a) |
| *Big- and meso-class rigidities* | | | | |
| 2a. 1b + meso-classes (Homogeneous) | 1417.4 | 335 | 0.034 | 0.000 (M.1a) |
| 2b. 1b + meso-classes (Heterogeneous) | 1415.6 | 332 | 0.034 | 0.615 (M.2a) |
| *Big-, meso-, micro-class rigidities* | | | | |
| 3a. 2a + micro-classes (Homogeneous) | 650.3 | 329 | 0.021 | 0.000 (M.2a) |
| 3b. 2a + micro-classes (Heterogeneous) | 645.1 | 323 | 0.021 | 0.636 (M.3a) |
| Panel B—adding self-employment and regulation dimensions to Panel A models | | | | |
| 4a. 2a + regulated prof. (Homogeneous) | 670.1 | 333 | 0.023 | 0.000 (M.2a) |
| 4b. 2a + regulated prof. (Heterogeneous) | 668.3 | 331 | 0.023 | 0.367 (M.4a) |
| *Big-, meso-class and self-employed professionals rigidities* | | | | |
| 5a. 2a + self-employed prof. (Homogeneous) | 649.3 | 333 | 0.023 | 0.000 (M.2a) |
| 5b. 2a + self-employed prof. (Heterogeneous) | 649.2 | 331 | 0.023 | 0.997 (M.5a) |
| *Big- and meso-class, regulated and self-employed prof. rigidities* | | | | |
| 6a. 2a + regulated x self-employed prof. (Homogeneous) | 477.7 | 331 | 0.017 | 0.000 (M.2a) |
| 6b. 2a + regulated x self-employed prof. (Heterogeneous) | 466.1 | 327 | 0.017 | 0.292 (M.6a) |
| *Big-, meso-, micro-class, + self-employed prof, (setting self-emp. and employed professionals per each micro-class)* | | | | |
| 7a. 3a + self-employed prof. (Homogeneous) | 418.5 | 317 | 0.016 | 0.000 (M.3a) |
| 7b. 3a + self-employed prof. (Heterogeneous) | 399.8 | 305 | 0.014 | 0.346 (M.7a) |

Data: SPL (2011).

Models 2a and 2b differentiate immobility within the Service class at the meso-level. Model 2a fits considerably better than 1a, thus suggesting that it is important to differentiate service class immobility at the meso-level. However, comparison of Models 2a and 2b suggests that at this level, gender differences in social immobility are not relevant: the fit of Model 2b is not significantly higher than Model 2a ($L^2 = 1417.4 - 1415.6 = 1.8$; df $= 335 - 332 = 3$; $p = 0.615$). Model 3a further differentiates micro-class rigidities within the meso-class of professionals while holding gender constant, whereas Model 3b allows for gender differences. Again, assuming micro-class differences improves model fits (comparing Models 2a and 2b with Models 3a and 3ab) when moving from Model 3a to 3b confirms that gender differences are significant ($L^2 = 650.3 - 645.1 = 5.2$; df $= 329 - 323 = 6$; $p = 0.636$).

This first set of models allowed us to test our first hypothesis. Our interpretation of these results is straightforward. First, the importance of micro-classes for explaining social immobility in the higher service class is evident, in line with previous research (e.g., Ruggera and Barone 2017). Second, the relevant differences in immobility by gender in Italy are related to the distinctions between occupations when moving from the big-class level to the meso classes (in line with Checchi 2010), whereas at both meso- and micro-class levels, social reproduction follows the same pattern for both men and women. To the best

of our knowledge, this is the first research comparing micro-class immobility at the top of the occupational hierarchy separately for men and women in Italy; however, for the origin-education association, Ruggera (2016) suggested lowering gender differences in professional employment by considering related fields of study at the higher tertiary education level. Similarities at the micro-class level across genders are illustrated in Figure 1. First, these results correspond with the main prediction of the social closure theory (cf. Parkin 1979). In Italy, social reproduction among micro-classes in regulated professions is stronger than that for unregulated professions, considering the social origin and education association at the tertiary education level (Ruggera 2021). Second, on analysing Italian higher education graduates (MA graduates in this context), gender differences tend to be less significant (Ruggera 2021; Schizzerotto 2002). Both sons and daughters of professionals have better chances of accessing regulated professions than children of other social classes, thus suggesting a higher level of gender equality for members of these professions (Weeden 2002; Murphy 1988). If the higher level of gender equality in these professions has been underlined in various studies (e.g., Pellizzari and Orsini 2012), specifically referring to the lower level of competition for members of licensed within professions, our study shows that advantages related to professional closure are reducing gender inequalities in the access. According to Weeden (2002) a higher level of gender equality within regulated professions emerges when analysing income inequalities at micro-level. In addition, Ruggera (2021, 2016) shows that gender differences in inequalities of educational opportunity by social class of origin decrease when the level of professional regulation increases. This author, indeed, analyses educational inequalities at horizontal level by considering fields of study in Italy. Hence, our study improve the existing literature by considering the intergenerational mobility of professionals micro-classes in Italy; and our results are in line with these studies, which considered educational inequalities as well as income inequalities by specifying a link between professional closure (translated in this context as the level of professional regulation) and weaker gender differences.

Figure 2 graphically shows the association between professional regulation and micro-class intergenerational immobility[8] by relating the index of the service market restrictiveness of each professional group as reported in Table 1 with beta coefficients extrapolated from Model 3b.

The results in Figure 2 are not meant to prove a genuine causal relationship between regulation and social dynamics, but they highlight that the association between regulation and intergenerational immobility is positive for both men and women, at least at the descriptive level. Essentially, Figure 2 suggests the main distinction between regulated and unrelated professions, which is tested in the next two models.

Models 4a and 4b only differentiate micro-class professionals according to whether the occupations are regulated or not, instead of differentiating them altogether from each other, as in Models 3a and 3b. This distinction improves the model fit significantly when compared to Models 2a and 4a (M2a vs. M4a: $L^2 = 1417.4 - 670.1 = 747.3$; df = 335 − 333 = 2; $p = 0.000$), assuming no differences beyond the meso-level classes in the higher service class. Again, gender differences do not improve the model fit (Model 4a vs. Model 4b). Thus, when testing our second hypothesis, beside Model 1b tested against Model 1a only with big-class, the log-linear analysis for social immobility shows that gender interaction is not statistically significant.

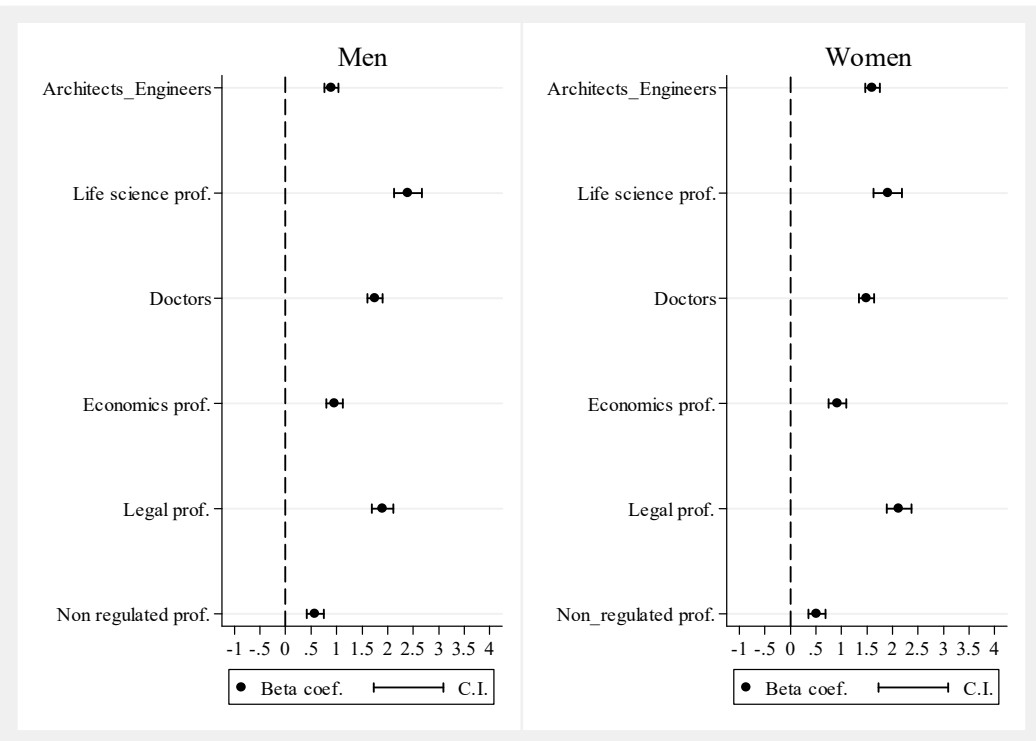

**Figure 1.** Micro-class parameters for social immobility in the higher service class across gender. Beta coefficients with confidence intervals extrapolated from Model 3b.

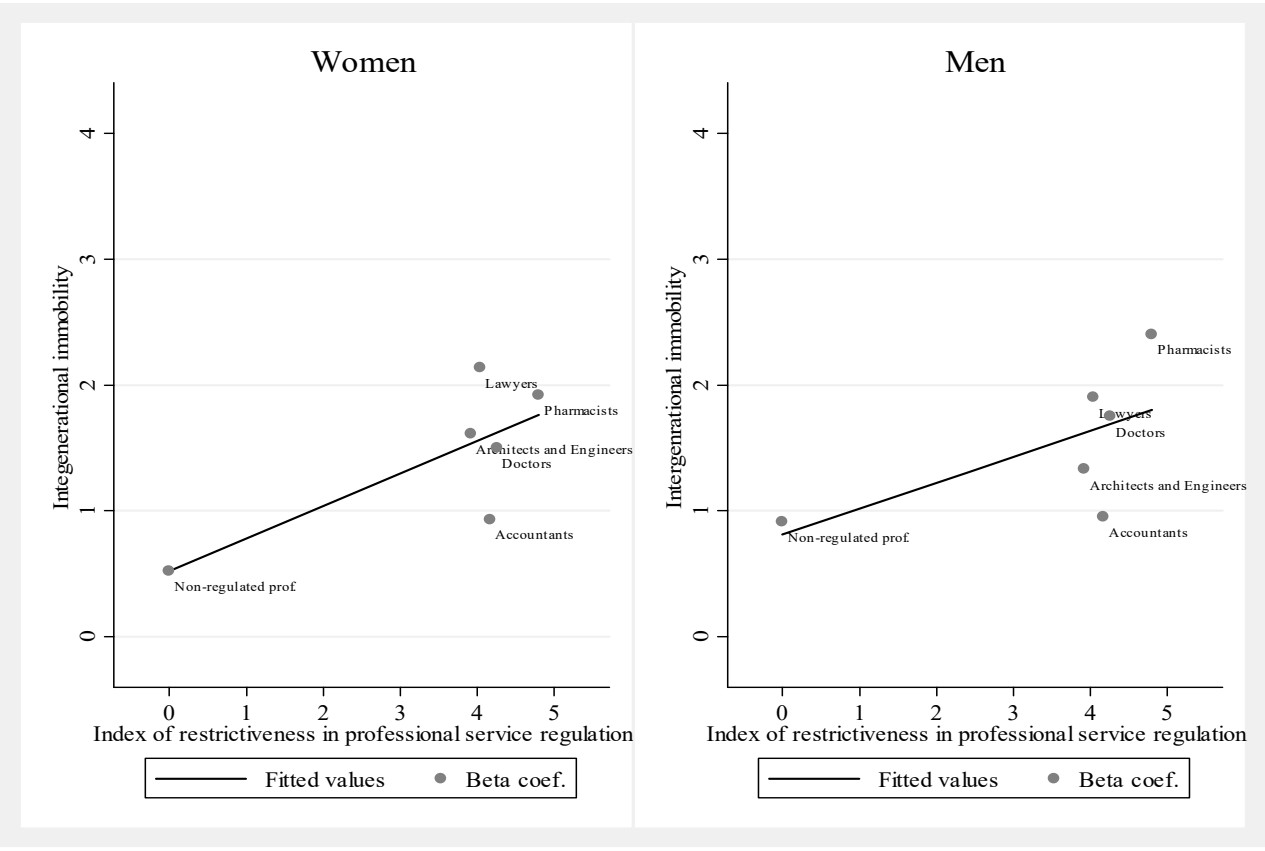

**Figure 2.** Correlation between entry market regulations in professional occupations (reported in Table 1) and immobility parameters for each micro-class extrapolated from model 3b.

Similarly, we considered the distinction of the micro-class occupations between self-employed and employed professionals. Likewise, regarding only meso-class distinctions, Models 5a and 5b fit significantly better than Models 2a and 2b (M2a vs. M5a: $L^2$ = 1417.4 − 649.3 = 767.8; df = 335 − 333 = 2; $p$ = 0.000). However, the improvement in the goodness of fit between Models 2a and 5a was slightly smaller than that of Models 2a and 4a, considering the importance of professional regulation. This leaves us with one option—to combine self-employment and regulation within the meso-class of professionals (2 × 2 dimension interaction) as in Models 6a and 6b (M2a vs. M6a: $L^2$ = 1417.4 − 649.3 = 915.5; df = 335 − 333 = 2; $p$ = 0.000). Our log-linear analysis concludes with two models concerning the role of self-employment within each micro-class (Models 7), which simply means that the occupations covered in Figure 1 are further differentiated by the employment situation (employed vs. self-employed professionals). In comparing Models 6a to Model 7a, we gain 59 of $L^2$ by losing 14 degrees of freedom, but the improvement is very poor when we focus on the dissimilarity index: **Δ** of model 6a is 0.017 and **Δ** of model 7a is 0.016. As already mentioned, Model 6a appears to be the model with the best fit after considering all indices. Model 7b, which also concerns gender differences, shows the best goodness of fit in the entire log-linear analysis. Moreover, Model 7a is accepted over Model 7b because gender differences are once again not significant, in line with hypothesis 2. These models allow treating each employed and self-employed professional group separately using the same reference category. This modelling strategy is particularly useful when considering social immobility at the top of the occupational hierarchy, whereas it could be less helpful if considering all diagonal cells into details. In other words, including lower classes specifications would employed too many parameters compromising the parsimony of the log-linear model.

Since Model 6a displays a higher significance in the goodness of fit model with respect to the previous ones (Models 4a and 5a), we show a clear graphical comparison between the results from Model 2a (at the top of the figure) on meso-class rigidities and Model 6a (at the bottom of the figure) in Figure 3.

The comparison of social class conceptualisations represented in Figure 3 shows a clear pattern in terms of meso-class rigidities. While larger entrepreneurs' intergenerational immobility is greater than that of professionals, the social reproduction of professionals is not significantly greater than that of higher managers. However, when differentiating professionals in terms of self-employment and regulation, the picture changes. Despite possessing the means to strengthen professionals' advantages from social closure, only self-employed professionals in regulated jobs achieve a similar level of immobility as larger entrepreneurs[9]. Later in the article, we show that self-employment related to professionals plays a stronger role to remain in the service class (I) than property for larger entrepreneurs (even when modelling the diagonal cells for immobility, property is crucial for larger entrepreneur's intergenerational immobility). We show that the higher the market regulation and professionals' specialised resources, the stronger the role played by self-employment.

To summarise our model selection, we consider $L^2$, degrees of freedom, and the dissimilarity of index, which is extremely informative, when choosing the best model. Comparing these three measures in model selection, Model 6a shows appropriate and parsimonious combinations of measures (especially considering Δ as 0.0017)[10]. Thus, we argue that based on the fit statistics of our log-linear analysis, Model 6a seems to be the best compromise. It shows the most parsimonious and effective way to understand the mechanisms of social immobility at the top of the occupational hierarchy in Italy, which is produced by the interaction of two crucial dimensions: professional entry-market regulation and self-employment.

A clarification at this point must be made. As reported in the Appendix A (on ILFI data analyses), meso-classes' immobility within the higher service class was calculated using two different designed surveys. The results obtained from the ILFI strictly correspond with our main analysis conducted on the large sample concerning only Italian graduates (SPL 2011). Specifically, we employ logistic regressions for both men and women by big and

meso class of origins (considering larger entrepreneurs, higher level managers, and high-level professionals) to calculate the propensity of being in the Service class (I). Afterwards, we consider only men using a micro-class approach, differentiating the micro-classes of professionals. We are extremely confident that analysing the OD association with ILFI data using individuals' first job allows for a comparison of the results for professionals at the beginning of their careers when using the SPL (2011). On the one hand, this confirms the robustness of our analyses among Italian graduates (for logistic analyses and professionals' micro-class immobility for men); on the other hand, using a specific survey on graduates provides us an opportunity to give adequate attention to women's social mobility, as well. Indeed, to the best of our knowledge, this is the first attempt in the Italian literature that considers both male and female graduates' labour market outcomes by different social class of origin schemes. Since the ILFI results are meant to confirm the robustness of the results from the SPL (2011), we report the whole analysis in Appendix A. However, we also believe that it will be more informative and clearer for the reader if the bivariate correlation between the level of professional regulation and micro-class immobility for men is shown from the ILFI data (using the same format as in Figure 2 for men and women—SPL 2011).

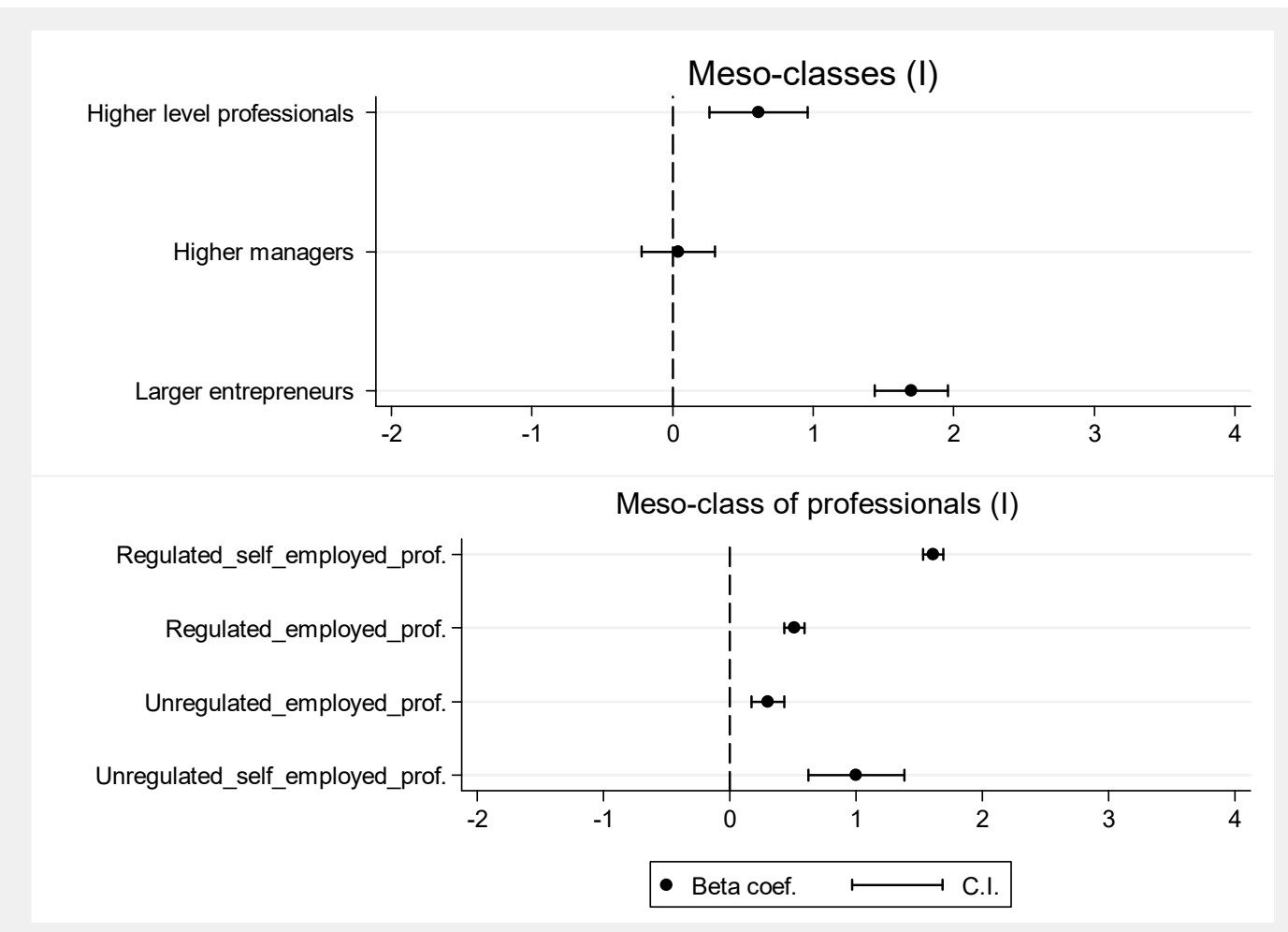

**Figure 3.** Beta parameters for social immobility for meso-classes and the additional interaction between self-employment and regulation at meso-level. Beta parameters with confidence intervals from Models 2a and 6a.

As Figure 4 shows, in terms of social micro-class immobility, regulated professions are distinguished from non-regulated professions even more evidently than men, as represented in Figure 2.[11]

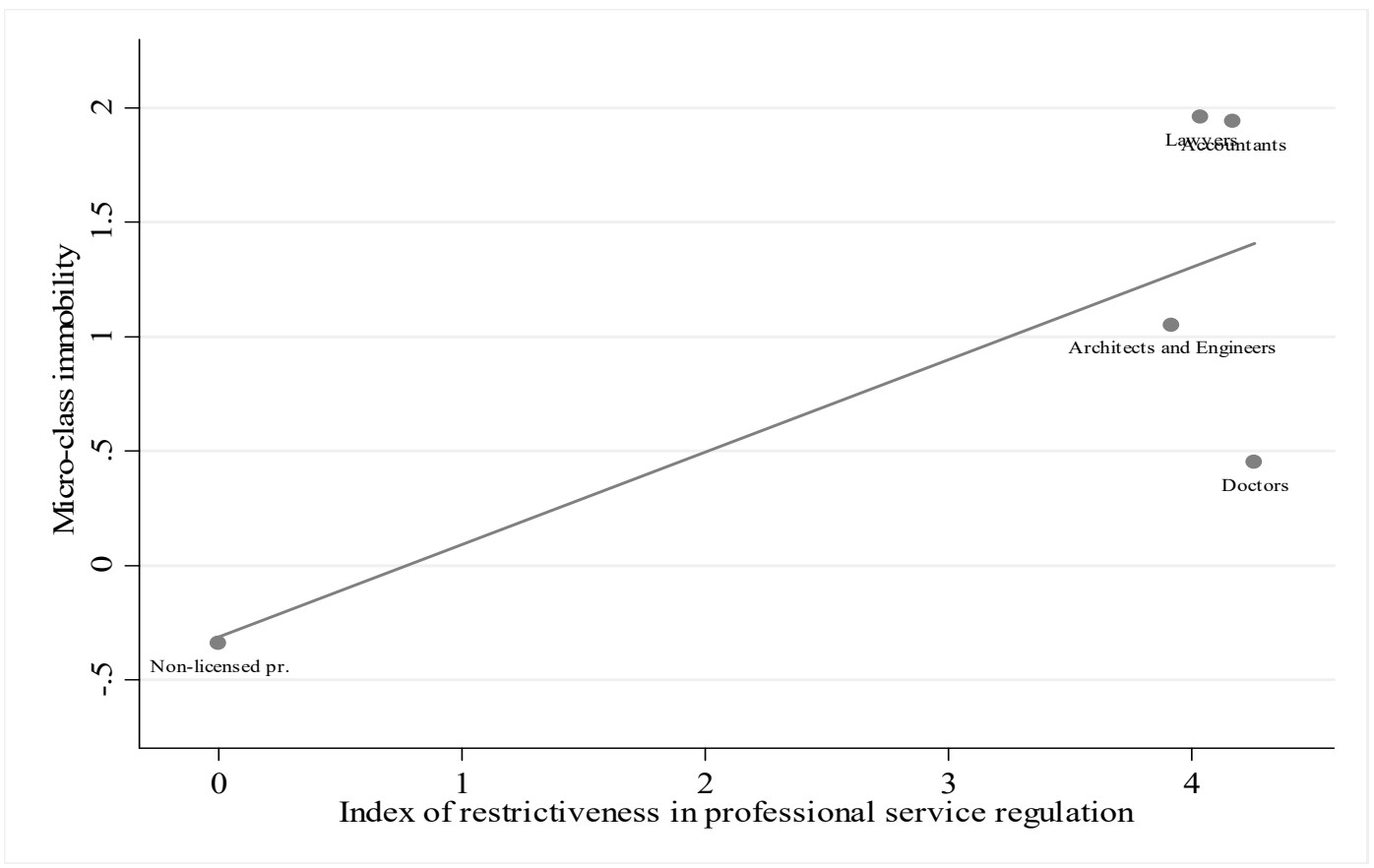

**Figure 4.** bivariate correlation between entry market regulations in professional occupations and immobility beta parameters for each micro-class for men extrapolated from Model 4 in Table A4 in the Appendix A). Source: ILFI survey (1997–2005). N. obs. 3840, men only.

## 6. Results: Gender Differences and Professional Children's Safety Nets for Social Immobility at the Top

To test our third hypothesis, we employed logistic regressions models using different specifications of social origins. We studied the roles of the meso- and micro-classes of origin as a safety net for the children of the upper class. In other words, we considered the chances of persistence at the top by also differentiating social origins in terms of self-employment. The results are graphically summarised in Figure 5.

First, we assessed big-class rigidities. Children of the higher service class displayed a greater likelihood of avoiding social demotion than other social classes; specifically, AME reached 15 and 18 percentage points for men and women, respectively. Second, we considered meso-classes within the higher service class; professionals' social immobility at the top is not significant with respect to the other meso-classes, which is in line with the results obtained with the ILFI data. For women, the AME of professionals are significant with respect to higher mangers. Third, we added two main distinctions in professional employment concerning regulation and self-employment dimensions. Our results show that safety nets work even better for daughters of regulated professionals than those of unregulated professionals, which is clearly observed from Figure 5 because the confidence intervals do not overlap in the case of women. When considering self-employed and employed professionals, AME are significant for both men and women, even if the gender differences are not significant for either dimension. We conclude the discussion of results presented in Figure 5 by considering self-employment within the upper-class fractions of regulated professionals.

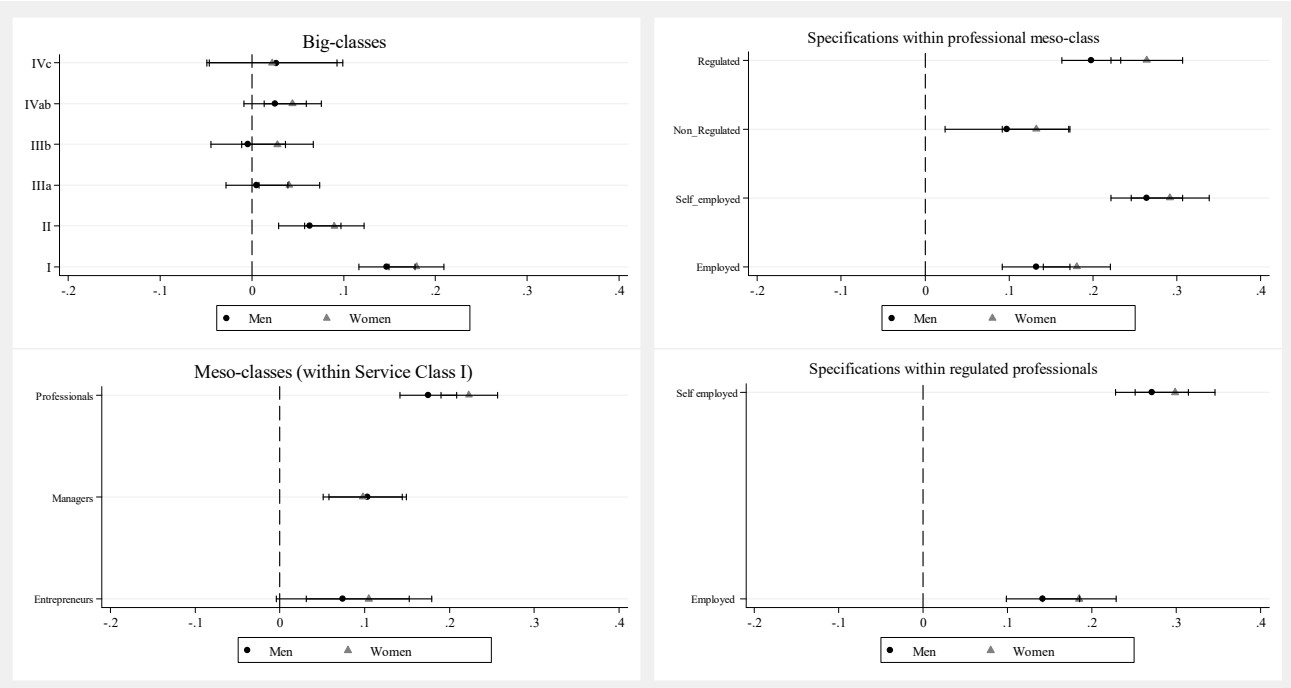

**Figure 5.** Average marginal effects from logistic regression models for the probability of being in the higher service class according to big-, meso-classes, and regulated employed and self-employed professionals (ref. cat. working-class V–VII). Estimates, separately for men and women, are at net of controls (age, secondary education grades, nationality, geographical area). Black dots denote the AME of social classes for men and the grey triangle for women, and black lines denote 95% confidence intervals. Legend: higher service class (I), lower service class (II), white collars (IIIa), routine non–manual workers (IIIb), petite bourgeoisie (IVab), farmers (IVc). Source: ISTAT- Sbocchi professionali dei laureati (SPL 2011).

These results are interesting, especially in terms of gender differences. On the one hand, the greatest chances of remaining in the higher service class concerns both sons and daughters of self-employed regulated professionals (AME 27 and 30 percentage points, respectively, for men and for women). As expected, it is higher than that of children of professionals in the meso-class. On the other hand, the combination of professional entry-market regulation and self-employment is more important for men than women. These results emerge from Figure 5 because, for women to remain within service class (I), having a regulated professional as a father is more significant than an unregulated professional. Similarly, it is more important to have a self-employed professional father than one who is merely employed. Instead, the regulation dimension itself is not significant for men; it only becomes significant when associated with self-employment.

Comparing the results from Figures 3 and 5, we derive two main considerations. First, the meso-class of professionals as a broader set of occupations shows a limited heuristic value in evaluating professions' mechanisms of social immobility at the top. This is because it does not allow the two distinctions in professional employment related to regulation and self-employment to be captured, which is the crucial combination to avoid social demotion (e.g., Barbagli and Schizzerotto 1997). Second, there are no significant gender differences in the functioning of safety nets when considering each category reported in Figure 5, except for the efficient role played by self-employment and regulation dimensions as discussed in the previous paragraphs. The results of the logistic regressions highlighting the different propensities of avoiding social demotions are very proficient in showing how heterogeneous the higher service class is, especially when considering processes of social closure, intergenerational immobility at the top, and gendered outcomes.

## 7. Conclusions

This article proves the fruitfulness of social closure predictions when explaining social immobility at the top of the occupational ladder. It shows that closure works through two main dimensions: self-employment, (i.e., size of client portfolio) and professional entry-market regulations. On the one hand, the processes of social reproduction come to light evidently when employing a non-standard approach to social mobility. The heuristic value of the micro-class proves to be useful in showing the correlation between regulation and professional social immobility, even if our results were only suggestive of genuine causal relationships; in fact, only bivariate association were detected. On the other hand, social closure beneficially addresses a higher level of gender equality within the micro-class with a stronger level of professional regulation, for which education, knowledge, and cognitive resources are more important for intergenerational immobility than such ascribed characteristics as gender.

Regulation in professional employment linked to self-employment enhances the economic rewards of members of professions, and thus, the specific transmission of parental resources takes place by closing off opportunities to outsiders. It has been proven that not only the children of the lower classes are outsiders but so are the children of unregulated professionals who are not provided with occupation-specific resources, such as cognitive, cultural, social, and fixed economic resources.

Implications for policy interventions can be easily linked to social exclusion and equalities of opportunity. Our considerations of this article's results relate to more equal regulation in professional employment, specifically regarding entry-market regulation. In the first place, incentives for children of the lower classes to enter highly regulated professions rise by lowering the costs of failure at the educational level and succeeding in the steps that regulate access to professions. Long and expansive steps articulate licensing systems after education. Thus, when occupation-specific resources transmitted from professional parents to children are in play, they incentivise these children to follow the winding path related to Italian licensed professions. From another point of view, if professions promote the use of credentials and licensing as main occupational closure strategies, they can also serve as a flywheel for gender equality. Social policy implications emerging from this study concern regulations that act as a lever for the promotion of gender egalitarian values but increase social exclusion for children from other social classes. Hence, since a general expansion of education is not included in the discussion, a more equal distribution of regulation in professional employment must be considered undoubtedly relevant to increase gender equality and decrease social inequality based on family resources.

Our suggestions for further research on this topic are not only to consider a micro-class approach to underlie processes of social exclusion and related implications for policy interventions and evaluations but also to consider regulation and self-employment as meso-class level specifications for both men and women. Even if micro-classes are beneficial in capturing professionals' social immobility, combining self-employment and regulation at the meso-class level improves the understanding of gender differences and captures processes of immobility at the top in a very parsimonious way. Finally, further studies should consider the role of mothers' occupational positions in detail (by considering mothers' versus daughters' mobility) by also employing other surveys from a comparative perspective, at least at the meso-class level and related specifications, as suggested in this article.

**Author Contributions:** Conceptualization, L.R. and J.E.; methodology, L.R.; software, J.E.; validation, L.R. and J.E.; formal analysis, L.R.; investigation, L.R.; resources, J.E.; data curation, J.E.; writing—original draft preparation, L.R.; writing—review and editing, J.E. All authors have read and agreed to the published version of the manuscript.

**Funding:** This work was supported by the INVEST research flagship under Grant [decision number 320162] and the Academy of Finland for NEFER under Grant [decision number 321264].

**Informed Consent Statement:** Informed consent was obtained from all subjects involved in the study.

**Data Availability Statement:** Not applicable.

**Conflicts of Interest:** The authors declare no conflict of interest.

## Appendix A

*Appendix A.1 Design Matrix for Different Approaches and Dimensions*

**Table A1.** Big-classes in Model 1a and 1b.

| | 1 | 2 | 3 | 4 | 5 | 6 | 7 | 8 | 9 | 10 | 11 | 12 | 13 | 14 | 15 | 16 | 17 |
|---|---|---|---|---|---|---|---|---|---|---|---|---|---|---|---|---|---|
| 1. Entrepreneurs | 1 | 1 | 1 | 1 | 1 | 1 | 1 | 1 | 1 | 1 | 1 | 1 | 1 | 1 | 5 | 6 | 7 |
| 2. Managers | 1 | 1 | 1 | 1 | 1 | 1 | 1 | 1 | 1 | 1 | 1 | 1 | 1 | 1 | 8 | 9 | 10 |
| 3. Arch. engin. empl. | 1 | 1 | 1 | 1 | 1 | 1 | 1 | 1 | 1 | 1 | 1 | 1 | 1 | 1 | 11 | 12 | 13 |
| 4. Arch. eng. self-emp. | 1 | 1 | 1 | 1 | 1 | 1 | 1 | 1 | 1 | 1 | 1 | 1 | 1 | 1 | 14 | 15 | 16 |
| 5. Life sc. empl. | 1 | 1 | 1 | 1 | 1 | 1 | 1 | 1 | 1 | 1 | 1 | 1 | 1 | 1 | 17 | 18 | 19 |
| 6. Life sc. self-emp | 1 | 1 | 1 | 1 | 1 | 1 | 1 | 1 | 1 | 1 | 1 | 1 | 1 | 1 | 20 | 21 | 22 |
| 7. Doctors emp | 1 | 1 | 1 | 1 | 1 | 1 | 1 | 1 | 1 | 1 | 1 | 1 | 1 | 1 | 23 | 24 | 25 |
| 8. Doctors self emp. | 1 | 1 | 1 | 1 | 1 | 1 | 1 | 1 | 1 | 1 | 1 | 1 | 1 | 1 | 26 | 27 | 28 |
| 9. Econ.prof. empl. | 1 | 1 | 1 | 1 | 1 | 1 | 1 | 1 | 1 | 1 | 1 | 1 | 1 | 1 | 29 | 30 | 31 |
| 10. Account. self-emp. | 1 | 1 | 1 | 1 | 1 | 1 | 1 | 1 | 1 | 1 | 1 | 1 | 1 | 1 | 32 | 33 | 34 |
| 11. Legal pr. emp. | 1 | 1 | 1 | 1 | 1 | 1 | 1 | 1 | 1 | 1 | 1 | 1 | 1 | 1 | 35 | 36 | 37 |
| 12. Legal pr. self-emp. | 1 | 1 | 1 | 1 | 1 | 1 | 1 | 1 | 1 | 1 | 1 | 1 | 1 | 1 | 38 | 39 | 40 |
| 13. unreg prof emp. | 1 | 1 | 1 | 1 | 1 | 1 | 1 | 1 | 1 | 1 | 1 | 1 | 1 | 1 | 41 | 42 | 43 |
| 14. unreg prof self-emp. | 1 | 1 | 1 | 1 | 1 | 1 | 1 | 1 | 1 | 1 | 1 | 1 | 1 | 1 | 44 | 45 | 46 |
| 15. II | 51 | 52 | 53 | 54 | 55 | 56 | 57 | 58 | 59 | 60 | 61 | 62 | 63 | 64 | 2 | 47 | 48 |
| 16. III | 65 | 66 | 67 | 68 | 69 | 70 | 71 | 72 | 73 | 74 | 75 | 76 | 77 | 78 | 79 | 3 | 49 |
| 17. IV–VII | 80 | 81 | 82 | 83 | 84 | 85 | 86 | 87 | 88 | 89 | 90 | 91 | 92 | 93 | 94 | 95 | 4 |

Legend: II: lower service class; III routine non-manual workers; IV–VII: self-employed and skilled unskilled manual workers.

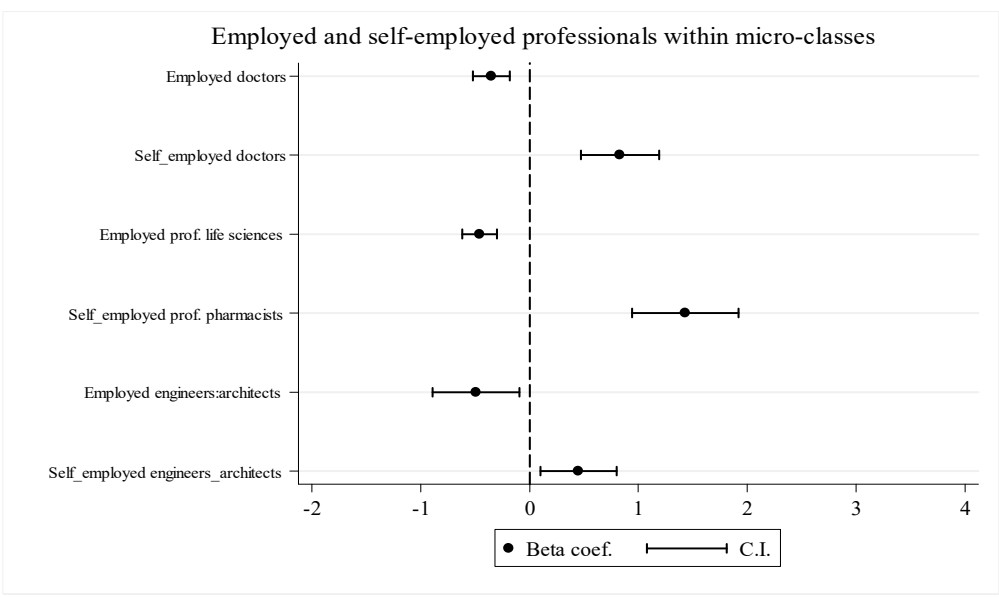

**Figure A1.** Beta parameters for micro-class social immobility specifying between self-employed and employed within each micro-class. Beta parameters with confidence intervals are extrapolated from Model 7a.

**Table A2.** Meso-class rigidities—Model 2a and 2b.

| | 1 | 2 | 3 | 4 | 5 | 6 | 7 | 8 | 9 | 10 | 11 | 12 | 13 | 14 | 15 | 16 | 17 |
|---|---|---|---|---|---|---|---|---|---|---|---|---|---|---|---|---|---|
| 1. Entrepreneurs | **2** | 1 | 1 | 1 | 1 | 1 | 1 | 1 | 1 | 1 | 1 | 1 | 1 | 1 | 1 | 1 | 1 |
| 2. Managers | 1 | **3** | 1 | 1 | 1 | 1 | 1 | 1 | 1 | 1 | 1 | 1 | 1 | 1 | 1 | 1 | 1 |
| 3. Arch. engin. empl. | 1 | 1 | **4** | **4** | **4** | **4** | **4** | **4** | **4** | **4** | **4** | **4** | **4** | **4** | 1 | 1 | 1 |
| 4. Arch. eng. self-emp. | 1 | 1 | **4** | **4** | **4** | **4** | **4** | **4** | **4** | **4** | **4** | **4** | **4** | **4** | 1 | 1 | 1 |
| 5. Life sc. empl. | 1 | 1 | **4** | **4** | **4** | **4** | **4** | **4** | **4** | **4** | **4** | **4** | **4** | **4** | 1 | 1 | 1 |
| 6. Life sc. self-emp | 1 | 1 | **4** | **4** | **4** | **4** | **4** | **4** | **4** | **4** | **4** | **4** | **4** | **4** | 1 | 1 | 1 |
| 7. Doctors emp | 1 | 1 | **4** | **4** | **4** | **4** | **4** | **4** | **4** | **4** | **4** | **4** | **4** | **4** | 1 | 1 | 1 |
| 8. Doctors self emp. | 1 | 1 | **4** | **4** | **4** | **4** | **4** | **4** | **4** | **4** | **4** | **4** | **4** | **4** | 1 | 1 | 1 |
| 9. Econ.prof. empl. | 1 | 1 | **4** | **4** | **4** | **4** | **4** | **4** | **4** | **4** | **4** | **4** | **4** | **4** | 1 | 1 | 1 |
| 10. Account. self-empl. | 1 | 1 | **4** | **4** | **4** | **4** | **4** | **4** | **4** | **4** | **4** | **4** | **4** | **4** | 1 | 1 | 1 |
| 11. Legal pr. emp. | 1 | 1 | **4** | **4** | **4** | **4** | **4** | **4** | **4** | **4** | **4** | **4** | **4** | **4** | 1 | 1 | 1 |
| 12. Legal pr. self-emp. | 1 | 1 | **4** | **4** | **4** | **4** | **4** | **4** | **4** | **4** | **4** | **4** | **4** | **4** | 1 | 1 | 1 |
| 13. unreg prof emp. | 1 | 1 | **4** | **4** | **4** | **4** | **4** | **4** | **4** | **4** | **4** | **4** | **4** | **4** | 1 | 1 | 1 |
| 14. unreg pr. self emp. | 1 | 1 | **4** | **4** | **4** | **4** | **4** | **4** | **4** | **4** | **4** | **4** | **4** | **4** | 1 | 1 | 1 |
| 15. II | 1 | 1 | 1 | 1 | 1 | 1 | 1 | 1 | 1 | 1 | 1 | 1 | 1 | 1 | 1 | 1 | 1 |
| 16. III | 1 | 1 | 1 | 1 | 1 | 1 | 1 | 1 | 1 | 1 | 1 | 1 | 1 | 1 | 1 | 1 | 1 |
| 17. IV–VII | 1 | 1 | 1 | 1 | 1 | 1 | 1 | 1 | 1 | 1 | 1 | 1 | 1 | 1 | 1 | 1 | 1 |

Legend: II: lower service class; III routine non-manual workers; IV–VII: self-employed and skilled unskilled manual workers.

**Table A3.** Meso-class rigidities—Model 3a and 3b.

| | 1 | 2 | 3 | 4 | 5 | 6 | 7 | 8 | 9 | 10 | 11 | 12 | 13 | 14 | 15 | 16 | 17 |
|---|---|---|---|---|---|---|---|---|---|---|---|---|---|---|---|---|---|
| 1. Entrepreneurs | 1 | 1 | 1 | 1 | 1 | 1 | 1 | 1 | 1 | 1 | 1 | 1 | 1 | 1 | 1 | 1 | 1 |
| 2. Managers | 1 | 1 | 1 | 1 | 1 | 1 | 1 | 1 | 1 | 1 | 1 | 1 | 1 | 1 | 1 | 1 | 1 |
| 3. Arch. engine. empl. | 1 | 1 | **2** | **2** | 1 | 1 | 1 | 1 | 1 | 1 | 1 | 1 | 1 | 1 | 1 | 1 | 1 |
| 4. Arch. eng. self-emp. | 1 | 1 | **2** | **2** | 1 | 1 | 1 | 1 | 1 | 1 | 1 | 1 | 1 | 1 | 1 | 1 | 1 |
| 5. Life sc. empl. | 1 | 1 | 1 | 1 | **3** | **3** | 1 | 1 | 1 | 1 | 1 | 1 | 1 | 1 | 1 | 1 | 1 |
| 6. Life sc. self-emp | 1 | 1 | 1 | 1 | **3** | **3** | 1 | 1 | 1 | 1 | 1 | 1 | 1 | 1 | 1 | 1 | 1 |
| 7. Doctors emp | 1 | 1 | 1 | 1 | 1 | 1 | **4** | **4** | 1 | 1 | 1 | 1 | 1 | 1 | 1 | 1 | 1 |
| 8. Doctors self-emp. | 1 | 1 | 1 | 1 | 1 | 1 | **4** | **4** | 1 | 1 | 1 | 1 | 1 | 1 | 1 | 1 | 1 |
| 9. Econ.prof. empl. | 1 | 1 | 1 | 1 | 1 | 1 | 1 | 1 | **5** | **5** | 1 | 1 | 1 | 1 | 1 | 1 | 1 |
| 10. Account. self-empl. | 1 | 1 | 1 | 1 | 1 | 1 | 1 | 1 | **5** | **5** | 1 | 1 | 1 | 1 | 1 | 1 | 1 |
| 11. Legal pr. emp. | 1 | 1 | 1 | 1 | 1 | 1 | 1 | 1 | 1 | 1 | **6** | **6** | 1 | 1 | 1 | 1 | 1 |
| 12. Legal pr. self-emp. | 1 | 1 | 1 | 1 | 1 | 1 | 1 | 1 | 1 | 1 | **6** | **6** | 1 | 1 | 1 | 1 | 1 |
| 13. Unreg .pr. emp. | 1 | 1 | 1 | 1 | 1 | 1 | 1 | 1 | 1 | 1 | 1 | 1 | **7** | **7** | 1 | 1 | 1 |
| 14. Unreg. pr. self-emp. | 1 | 1 | 1 | 1 | 1 | 1 | 1 | 1 | 1 | 1 | 1 | 1 | **7** | **7** | 1 | 1 | 1 |
| 15. II | 1 | 1 | 1 | 1 | 1 | 1 | 1 | 1 | 1 | 1 | 1 | 1 | 1 | 1 | 1 | 1 | 1 |
| 16. III | 1 | 1 | 1 | 1 | 1 | 1 | 1 | 1 | 1 | 1 | 1 | 1 | 1 | 1 | 1 | 1 | 1 |
| 17. IV–VII | 1 | 1 | 1 | 1 | 1 | 1 | 1 | 1 | 1 | 1 | 1 | 1 | 1 | 1 | 1 | 1 | 1 |

Legend: II: lower service class; III routine non-manual workers; IV–VII: self-employed and skilled unskilled manual workers.

**Table A4.** Regulated and unregulated professionals—Model 4a and 4b.

| | 1 | 2 | 3 | 4 | 5 | 6 | 7 | 8 | 9 | 10 | 11 | 12 | 13 | 14 | 15 | 16 | 17 |
|---|---|---|---|---|---|---|---|---|---|---|---|---|---|---|---|---|---|
| 1. Entrepreneurs | 1 | 1 | 1 | 1 | 1 | 1 | 1 | 1 | 1 | 1 | 1 | 1 | 1 | 1 | 1 | 1 | 1 |
| 2. Managers | 1 | 1 | 1 | 1 | 1 | 1 | 1 | 1 | 1 | 1 | 1 | 1 | 1 | 1 | 1 | 1 | 1 |
| 3. Arch. engine. empl. | 1 | 1 | **2** | **2** | 1 | 1 | 1 | 1 | 1 | 1 | 1 | 1 | 1 | 1 | 1 | 1 | 1 |
| 4. Arch. eng. self-emp. | 1 | 1 | **2** | **2** | 1 | 1 | 1 | 1 | 1 | 1 | 1 | 1 | 1 | 1 | 1 | 1 | 1 |
| 5. Life sc. empl. | 1 | 1 | 1 | 1 | **2** | **2** | 1 | 1 | 1 | 1 | 1 | 1 | 1 | 1 | 1 | 1 | 1 |
| 6. Life sc. self-emp | 1 | 1 | 1 | 1 | **2** | **2** | 1 | 1 | 1 | 1 | 1 | 1 | 1 | 1 | 1 | 1 | 1 |
| 7. Doctors emp | 1 | 1 | 1 | 1 | 1 | 1 | **2** | **2** | 1 | 1 | 1 | 1 | 1 | 1 | 1 | 1 | 1 |
| 8. Doctors self-emp. | 1 | 1 | 1 | 1 | 1 | 1 | **2** | **2** | 1 | 1 | 1 | 1 | 1 | 1 | 1 | 1 | 1 |
| 9. Econ.prof. empl. | 1 | 1 | 1 | 1 | 1 | 1 | 1 | 1 | **2** | **2** | 1 | 1 | 1 | 1 | 1 | 1 | 1 |
| 10. Account. self-empl. | 1 | 1 | 1 | 1 | 1 | 1 | 1 | 1 | **2** | **2** | 1 | 1 | 1 | 1 | 1 | 1 | 1 |
| 11. Legal pr. emp. | 1 | 1 | 1 | 1 | 1 | 1 | 1 | 1 | 1 | 1 | **2** | **2** | 1 | 1 | 1 | 1 | 1 |
| 12. Legal pr. self-emp. | 1 | 1 | 1 | 1 | 1 | 1 | 1 | 1 | 1 | 1 | **2** | **2** | 1 | 1 | 1 | 1 | 1 |
| 13. Unreg .pr. emp. | 1 | 1 | 1 | 1 | 1 | 1 | 1 | 1 | 1 | 1 | 1 | 1 | **3** | **3** | 1 | 1 | 1 |
| 14. Unreg. pr. self-emp. | 1 | 1 | 1 | 1 | 1 | 1 | 1 | 1 | 1 | 1 | 1 | 1 | **3** | **3** | 1 | 1 | 1 |
| 15. II | 1 | 1 | 1 | 1 | 1 | 1 | 1 | 1 | 1 | 1 | 1 | 1 | 1 | 1 | 1 | 1 | 1 |
| 16. III | 1 | 1 | 1 | 1 | 1 | 1 | 1 | 1 | 1 | 1 | 1 | 1 | 1 | 1 | 1 | 1 | 1 |
| 17. IV–VII | 1 | 1 | 1 | 1 | 1 | 1 | 1 | 1 | 1 | 1 | 1 | 1 | 1 | 1 | 1 | 1 | 1 |

Legend: II: lower service class; III routine non-manual workers; IV–VII: self-employed and skilled unskilled manual workers.

**Table A5.** Employed and self-employed professionals—Model 5a and 5b.

| | 1 | 2 | 3 | 4 | 5 | 6 | 7 | 8 | 9 | 10 | 11 | 12 | 13 | 14 | 15 | 16 | 17 |
|---|---|---|---|---|---|---|---|---|---|---|---|---|---|---|---|---|---|
| 1. Entrepreneurs | 1 | 1 | 1 | 1 | 1 | 1 | 1 | 1 | 1 | 1 | 1 | 1 | 1 | 1 | 1 | 1 | 1 |
| 2. Managers | 1 | 1 | 1 | 1 | 1 | 1 | 1 | 1 | 1 | 1 | 1 | 1 | 1 | 1 | 1 | 1 | 1 |
| 3. Arch. engin. empl. | 1 | 1 | **2** | 1 | 1 | 1 | 1 | 1 | 1 | 1 | 1 | 1 | 1 | 1 | 1 | 1 | 1 |
| 4. Arch. eng. self-emp. | 1 | 1 | 1 | **3** | 1 | 1 | 1 | 1 | 1 | 1 | 1 | 1 | 1 | 1 | 1 | 1 | 1 |
| 5. Life sc. empl. | 1 | 1 | 1 | 1 | **2** | 1 | 1 | 1 | 1 | 1 | 1 | 1 | 1 | 1 | 1 | 1 | 1 |
| 6. Life sc. self-emp | 1 | 1 | 1 | 1 | 1 | **3** | 1 | 1 | 1 | 1 | 1 | 1 | 1 | 1 | 1 | 1 | 1 |
| 7. Doctors emp | 1 | 1 | 1 | 1 | 1 | 1 | **2** | 1 | 1 | 1 | 1 | 1 | 1 | 1 | 1 | 1 | 1 |
| 8. Doctors self emp. | 1 | 1 | 1 | 1 | 1 | 1 | 1 | **3** | 1 | 1 | 1 | 1 | 1 | 1 | 1 | 1 | 1 |
| 9. Econ.prof. empl. | 1 | 1 | 1 | 1 | 1 | 1 | 1 | 1 | **2** | 1 | 1 | 1 | 1 | 1 | 1 | 1 | 1 |
| 10. Account. self-em. | 1 | 1 | 1 | 1 | 1 | 1 | 1 | 1 | 1 | **3** | 1 | 1 | 1 | 1 | 1 | 1 | 1 |
| 11. Legal pr. emp. | 1 | 1 | 1 | 1 | 1 | 1 | 1 | 1 | 1 | 1 | **2** | 1 | 1 | 1 | 1 | 1 | 1 |
| 12. Legal pr. self-emp. | 1 | 1 | 1 | 1 | 1 | 1 | 1 | 1 | 1 | 1 | 1 | **3** | 1 | 1 | 1 | 1 | 1 |
| 13. Unreg prof emp. | 1 | 1 | 1 | 1 | 1 | 1 | 1 | 1 | 1 | 1 | 1 | 1 | **2** | 1 | 1 | 1 | 1 |
| 14. Unreg. pr. self-emp. | 1 | 1 | 1 | 1 | 1 | 1 | 1 | 1 | 1 | 1 | 1 | 1 | 1 | **3** | 1 | 1 | 1 |
| 15. II | 1 | 1 | 1 | 1 | 1 | 1 | 1 | 1 | 1 | 1 | 1 | 1 | 1 | 1 | 1 | 1 | 1 |
| 16. III | 1 | 1 | 1 | 1 | 1 | 1 | 1 | 1 | 1 | 1 | 1 | 1 | 1 | 1 | 1 | 1 | 1 |
| 17. IV–VII | 1 | 1 | 1 | 1 | 1 | 1 | 1 | 1 | 1 | 1 | 1 | 1 | 1 | 1 | 1 | 1 | 1 |

Legend: II: lower service class; III routine non-manual workers; IV–VII self-employed workers and skilled unskilled manual workers.

**Table A6.** Employed and self-employed professionals—Model 5a and 5b.

| | 1 | 2 | 3 | 4 | 5 | 6 | 7 | 8 | 9 | 10 | 11 | 12 | 13 | 14 | 15 | 16 | 17 |
|---|---|---|---|---|---|---|---|---|---|---|---|---|---|---|---|---|---|
| 1. Entrepreneurs | 1 | 1 | 1 | 1 | 1 | 1 | 1 | 1 | 1 | 1 | 1 | 1 | 1 | 1 | 1 | 1 | 1 |
| 2. Managers | 1 | 1 | 1 | 1 | 1 | 1 | 1 | 1 | 1 | 1 | 1 | 1 | 1 | 1 | 1 | 1 | 1 |
| 3. Arch. engin. empl. | 1 | 1 | **2** | 1 | 1 | 1 | 1 | 1 | 1 | 1 | 1 | 1 | 1 | 1 | 1 | 1 | 1 |
| 4. Arch. eng. self-emp. | 1 | 1 | 1 | **3** | 1 | 1 | 1 | 1 | 1 | 1 | 1 | 1 | 1 | 1 | 1 | 1 | 1 |
| 5. Life sc. empl. | 1 | 1 | 1 | 1 | **4** | 1 | 1 | 1 | 1 | 1 | 1 | 1 | 1 | 1 | 1 | 1 | 1 |
| 6. Life sc. self-emp | 1 | 1 | 1 | 1 | 1 | **5** | 1 | 1 | 1 | 1 | 1 | 1 | 1 | 1 | 1 | 1 | 1 |
| 7. Doctors emp | 1 | 1 | 1 | 1 | 1 | 1 | **6** | 1 | 1 | 1 | 1 | 1 | 1 | 1 | 1 | 1 | 1 |
| 8. Doctors self emp. | 1 | 1 | 1 | 1 | 1 | 1 | 1 | **7** | 1 | 1 | 1 | 1 | 1 | 1 | 1 | 1 | 1 |
| 9. Econ. prof. empl. | 1 | 1 | 1 | 1 | 1 | 1 | 1 | 1 | **8** | 1 | 1 | 1 | 1 | 1 | 1 | 1 | 1 |
| 10. Account. self-emp. | 1 | 1 | 1 | 1 | 1 | 1 | 1 | 1 | 1 | **9** | 1 | 1 | 1 | 1 | 1 | 1 | 1 |
| 11. Legal pr. emp. | 1 | 1 | 1 | 1 | 1 | 1 | 1 | 1 | 1 | 1 | **10** | 1 | 1 | 1 | 1 | 1 | 1 |
| 12. Legal pr. self-emp. | 1 | 1 | 1 | 1 | 1 | 1 | 1 | 1 | 1 | 1 | 1 | **11** | 1 | 1 | 1 | 1 | 1 |
| 13. Unreg prof emp. | 1 | 1 | 1 | 1 | 1 | 1 | 1 | 1 | 1 | 1 | 1 | 1 | **12** | 1 | 1 | 1 | 1 |
| 14. Unreg pr. self-emp. | 1 | 1 | 1 | 1 | 1 | 1 | 1 | 1 | 1 | 1 | 1 | 1 | 1 | **13** | 1 | 1 | 1 |
| 15. II | 1 | 1 | 1 | 1 | 1 | 1 | 1 | 1 | 1 | 1 | 1 | 1 | 1 | 1 | 1 | 1 | 1 |
| 16. III | 1 | 1 | 1 | 1 | 1 | 1 | 1 | 1 | 1 | 1 | 1 | 1 | 1 | 1 | 1 | 1 | 1 |
| 17. IV–VII | 1 | 1 | 1 | 1 | 1 | 1 | 1 | 1 | 1 | 1 | 1 | 1 | 1 | 1 | 1 | 1 | 1 |

Legend: II: lower service class; III routine non-manual workers; IV–VII self-employed workers and skilled unskilled manual workers.

*Appendix A.2 Analysis C.1 with ILFI Data*

Figure A2 shows results for the propensity of being in the higher service class according to big- and meso-classes. We do not present results for men and women separately because they do not present significant differences. Controls concern cohort of birth, geographical areas, gender, education, and nationality.

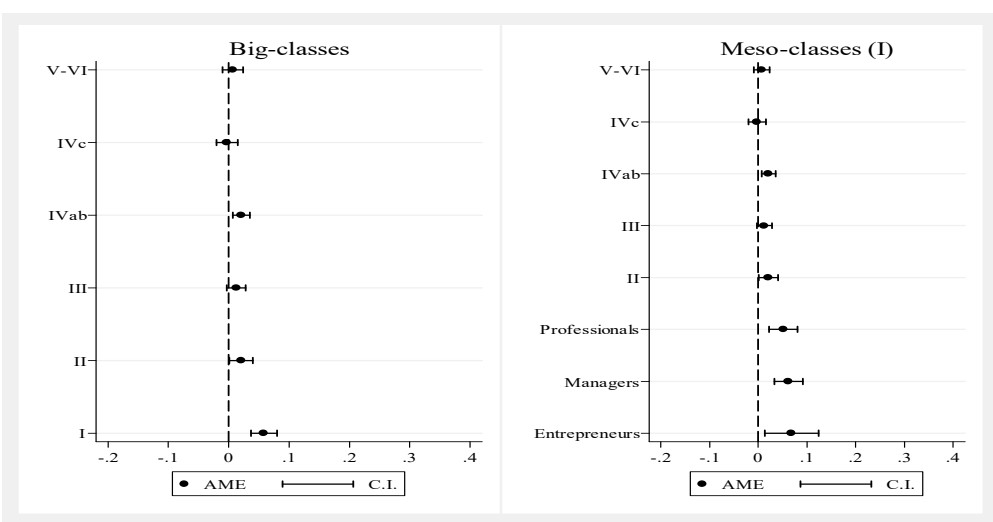

**Figure A2.** Average marginal effects from logistic regression models for the probability of being in the higher service class according to big- and meso-classes (ref. cat. working-class VII). Estimates are at net of controls (gender, geographical areas, cohort of birth, and education). Black dots denote the AME of social classes for men, and the gray triangle for women and black lines denote 95% confidence intervals. Legend: higher service class (I), lower service class (II), routine non–manual workers (III), petite bourgeoisie (IVab), farmers (IVc), skilled manual workers (V–VI). Source: ILFI (1997–2005).

**Table A7.** Fit indices of log-linear model of quasi perfect mobility with Big-, Meso-, and Micro-class rigidities (men only 3840 valid cases).

| Model Description | $L^2$ | d.f. | Δ | Significance |
|---|---|---|---|---|
| 0. Independence Model | 1938.07 | 144 | 0.180 | |
| 1. Big-class rigidities | 967.01 | 137 | 0.105 | (M0. 0.000) |
| 2. Big- and -meso class rigidities | 934.88 | 134 | 0.104 | (M1. 0.000) |
| 3. Big- meso- and regulated professions | 927.47 | 132 | 0.104 | (M1. 0.000) (M2. 0.018) |
| 4. Big-, meso- and micro-class rigidities | 923.10 | 129 | 0.103 | (M1. 0.000) (M2. 0.014) |

Source: ILFI (1997–2005).

The baseline model of conditional independence is a basic model that unrealistically supposed a missed relationship between social origins and destinations. Instead, Models 1a and 1b are a yardstick for the comparison with other models pertaining to immobility within the higher service class. Model 1 only adds big-class rigidities and Model 2 adds three additional parameters capturing meso-class rigidities within the higher service class. Models 3 and 4 incorporate immobility mechanisms for regulated professionals and micro-classes. The improvement in the goodness of fit highlights three main conclusions. First, the heuristic value of the standard approach to social immobility is certainly relevant; second, meso-classes are useful to further capture immobility at the top; third, the micro-classes of professionals are slightly significant (*p*-value < 0.05), and distinction between regulated and unregulated professionals can show the differences among professionals. However, even if the reduction of $L^2$ for micro-classes is not huge, at a pure descriptive level, the association between regulation and the social reproduction of professional groups (reported in Figure 4 main text) is positive and in line with our main analysis on Italian graduates.

## Notes

[1] In this context, the term licensed professions refers to highly regulated professions that are organized in professional associations; and that require a mandatory license in order to practice a given set of exclusive tasks (i.e., lawyers, notaries, doctors, architects, engineers).

[2] Since engineers and architects display the same level of entry-market regulation, they can be classified, more generally, as technical licensed professions.

[3] Main distributions of frequencies are available upon request.

[4] According to ILFI data among graduates, with regards their fathers, more than 85% are accountants in the professional group of economics, more than 70% are pharmacists and veterinarians in the life science group, and more than 75% are lawyers, judges, and notaries in the legal profession group. When considering self-employed professionals, all of them are licensed, which is defined as liberal professionals in the survey.

[5] According to ILFI data among graduates, less than 8% are psychologist fathers in the micro-class, which is not representative of this professional group. Moreover, they have been represented by a professional association within a licensure system since 1989; the level of regulation is not high because they do not perform any exclusive tasks (only psychiatrists perform tasks with exclusivity).

[6] CP2011 is ISTAT occupational code, which differs from ISCO88 only because it provides higher-level occupational specifications. Thus, the three-digits occupational code integrated with the other information on professionals allows us to have more confidence in the micro-class operationalisation.

[7] Since we rely on occupational codes first before licenses, which tell us who defined themselves as, for example, doctors, architects, or engineers (85% to 95% of the self-employed already obtained a license, while the others are employed in programs such as 'praticantato') in order the professional license. Given these high percentage misclassifications cannot be consider an issue in this work. Moreover, we also repeat the log-linear analysis with ILFI data, a survey representative of the entire population and micro-class immobility of accountants is undeniable high.

[8] To facilitate the reader's understanding, we use the same denominations of micro-class categories in the graphs as in Table 1 as well to compare the results of Figure 2 with those of Figure 5.

9    Considering each diagonal cells for social immobility, these results can change with the outcome variable. By using a logistic regression with a two category variable (being in the Service class I against all the other categories), the parameters can assume different magnitudes, including those regarding larger entrepreneurs compared to professionals, which have various resources to employ more easily in other service class occupations.

10   Figure A1, which shows some examples of the further distinctions of the micro-classes extrapolated from Model 7a, is reported in the Appendix A.

11   One objection that could arise when analysing (MA) graduates against a survey on entire populations is an overestimation of the social immobility effect. Figure 4 shows that this is not the case. Once again, we assert that the SPL (2011) results are robust.

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
