# Peer review of "Licensed Professionals and Intergenerational Big-, Meso- and Micro-Class Immobility within the Upper Class; Social Closure and Gendered Outcomes among Italian Graduates"

_socsci, doi:10.3390/socsci11090418_

Round 1

Reviewer 1 Report

The article studies how social closure influences intergenerational immobility at the top occupation in Italy. In the paper, the authors use two different survey data and log-linear modeling as well as logistic regression. The SPL sample that the authors use allows them to study social mobility at the beginning of their career which can be considered a merit of the dataset. The topic of the study is interesting, how social closure of the occupation influence intergenerational social mobility. Usually, studies have estimated how occupational closure influences for example income mobility or unemployment, thus I think this is a novel approach to the topic. Also, the large main sample (over 19.554) and those results have been replicated with the other sample can be considered as a merit of the paper. Loglinear modeling as a method is not used often in social sciences anymore but for this paper, it seems a wise choice because measured occupational variables are categorical. The micro class approach is also interesting and gives a more detailed picture of intergenerational occupation mobility and can be considered a new contribution to the literature. My further comments are below. 

Because the micro class approach is rather a new way of analyzing occupations, I think it should be explained in a more detailed manner in the introduction. 

Sometimes authors use (for example in the Introduction) the word social mobility to describe the association between parents and children, however, for me, social mobility means individual mobility. Authors should use the term intergenerational social mobility to distinguish individual social mobility from intergenerational social mobility. 

In the theoretical background authors only discuss the socialization of the children in parental occupations. However, there are now available new studies on the genetic inheritance of occupations showing that genetic endowments can be more important than socialization or cultural aspects of the family (see Erola, Lehti, Baier & Karhula 2022). Authors should add this discussion to the theoretical background and cite relevant studies. Further, authors should not assume that for example cognitive is divided evenly in the population. There are differences and these differences are reflected in intergenerational social mobility through education. Therefore, the beginning of the theoretical background should be framed differently and begin with endowments and investments that parents transmit to children that are relevant to occupational mobility. The role of education should also be discussed more comprehensive way.    

Authors write about class-specific socialization of the women and men and different expectations for men and women, however, there are now some findings that socialization does not explain the difference between education and occupational choices of men and women, because in more free countries men and women occupation (and education) fields are even more stratified. This is called the gender inequality paradox. Thus, my suggestion is to reduce discussion on male and female socialization or frame this part differently. 

Further, I did not find any explicitly written hypothesis on how family background influences gender differences in occupation mobility. Authors should add the hypothesis. The other option would be just dropping the discussion on gender; however, I think that is not a wise decision because women's and men's occupational paths differ, and it would be interesting to see their differences according to parental background. 

In many parts, the authors write that the micro classes are mechanisms, but I did not understand how this micro class mechanism functions. Authors should clarify this in the theoretical background.  

The authors discuss at the beginning of Section 3 on occupation and education systems of Nordic countries and Germany. I think the authors should describe in a more detailed manner Italian labor markets and education systems before comparing them to other countries. The beginning of this section is somewhat confusing because it does not mention table 1 it might refer to. Thus, it would be better first to describe the Italian system and then compare it to other countries. However, I think table 1 is a nice presentation of the labor market closure of the different occupations between the countries.    

I think authors should explain even better their target population. Who are the Italian graduates? You write that “we do not consider those graduates who only hold a BA degree”, thus my question is who are the graduates? You should clarify the population under the study. Also, if you only follow graduates from higher education your results cannot be generalized to the whole population. Thus, you should address that the study population is higher educated and their parents. 

At least in the logistic regressions, many control variables have been used. Authors should explain, what these control variables are in the data and methods section. Further, they should add the description of logistic regression and AME. I mean what are the models that you use. Please, clarify this. 

Minor comments 

It would be more reader-friendly to separate the method section from the data section. 

Can you clarify your figure 5 caption, is the y-axis always parental occupational standing?  

There is a typo in the 5. title in the word gender. 

References 

Erola, J., Lehti, H., Baier, T., & Karhula, A. (2022). Socioeconomic Background and Gene-Environment Interplay in Social Stratification across the Early Life Course. European Sociological Review38(1), 1-17.

Author Response

Response to Reviewer 1 Comments

Before proceeding with my responses to the comments, I would like to thank the reviewers for the helpful suggestions and comments.

The article studies how social closure influences intergenerational immobility at the top occupation in Italy. In the paper, the authors use two different survey data and log-linear modeling as well as logistic regression. The SPL sample that the authors use allows them to study social mobility at the beginning of their career which can be considered a merit of the dataset. The topic of the study is interesting, how social closure of the occupation influence intergenerational social mobility. Usually, studies have estimated how occupational closure influences for example income mobility or unemployment, thus I think this is a novel approach to the topic. Also, the large main sample (over 19.554) and those results have been replicated with the other sample can be considered as a merit of the paper. Loglinear modeling as a method is not used often in social sciences anymore but for this paper, it seems a wise choice because measured occupational variables are categorical. The micro class approach is also interesting and gives a more detailed picture of intergenerational occupation mobility and can be considered a new contribution to the literature. My further comments are below.

Point 1: Because the micro class approach is rather a new way of analyzing occupations, I think it should : be explained in a more detailed manner in the introduction.

Response 1: we agree. As in the introduction – line 34 footnote – when we mention the micro-class approach we add a specific definition of micro-classes in this context (see the first footnote of the article).

Point 2: Sometimes authors use (for example in the Introduction) the word social mobility to describe the association between parents and children, however, for me, social mobility means individual mobility. Authors should use the term intergenerational social mobility to distinguish individual social mobility from intergenerational social mobility.

Response 2: this is true, social mobility and intergenerational social mobility refer to different processes. And in the introduction, since in the first line we properly say that we study integenrational social mobility we leave social mobility right after, because we avoid repetitions, but we changed social mobility with integenerational social mobility in the middle of the introduction Line 50.

Point 3: In the theoretical background authors only discuss the socialization of the children in parental occupations. However, there are now available new studies on the genetic inheritance of occupations showing that genetic endowments can be more important than socialization or cultural aspects of the family (see Erola, Lehti, Baier & Karhula 2022). Authors should add this discussion to the theoretical background and cite relevant studies. Further, authors should not assume that for example cognitive is divided evenly in the population. There are differences and these differences are reflected in intergenerational social mobility through education. Therefore, the beginning of the theoretical background should be framed differently and begin with endowments and investments that parents transmit to children that are relevant to occupational mobility. The role of education should also be discussed more comprehensive way.    

Response 3: literature on genetic endowments is very interesting and as suggested we add the reference as Erola et al. 2022 – see line 88. However, despite the literature on the transmission of cognitive skills is vast in this article we decided to put the attention on the four different capitals transmitted from parents to children at different level – big-, meso- and micro-level. This is to help the reader to understand  different approaches employed in the analysis and why we need to employ them. Then, we decided to focus on intergenerational social mobility and micro-level mechanisms (in the Coleman boat) as rational choices or when children are pushed from behind – according to Bourdieu for example. Our choice was to drive the reader into a more specific literature, as the reviewer noticed, and to avoid other sectors which, even if very interesting, could be a bit confusing. However, since the reference has been added the reader can decide to go into details of the suggested literature as well.

The role of education in the regulated Italian profession are explain when we discuss professional closure in Italy. For example, entry requirements are MA degrees in specific field of study – see for example the case of medical doctors.

Point 4: Authors write about class-specific socialization of the women and men and different expectations for men and women, however, there are now some findings that socialization does not explain the difference between education and occupational choices of men and women, because in more free countries men and women occupation (and education) fields are even more stratified. This is called the gender inequality paradox. Thus, my suggestion is to reduce discussion on male and female socialization or frame this part differently.

Further, I did not find any explicitly written hypothesis on how family background influences gender differences in occupation mobility. Authors should add the hypothesis. The other option would be just dropping the discussion on gender; however, I think that is not a wise decision because women's and men's occupational paths differ, and it would be interesting to see their differences according to parental background.

Response 4: this point is strictly related with the previous one on education. By distinguishing within the meso-class of professionals in the upper class (big - class) we give attention to regulated and non regulated professions, employed and self-employed professionals – all this is done for men and women separately. As the analysis confirm when focusing on this professions men and  women have the same incentives to follow into their fathers footsteps. We thank the reviewer for the attention to this point, but we did not expect big differences in this case (as Weeden or Barone already pointed out).

Furthermore, it emerges that gender differences are more related to gender composition of the educational field of study as in the case of technical profession as discussed in the discussion of results. Otherwise, results are in line with our hypotheses and with Weeden and Grusky or Barone, Ruggera and Barone about incentive related to highy regulated profession. We believe that dropping the discussion on gender does not allow the reader to understand the gender similarity more probably related to strong incentive related the level of regulation. We suggest to deepen this kind of hypotheses in future research by considering regulated professions and the mothers´occupations (with other data or/and also in other countries).

Point 5: In many parts, the authors write that the micro classes are mechanisms, but I did not understand how this micro class mechanism functions. Authors should clarify this in the theoretical background. 

Response 5: please see the points clarified before on the three different approach employed. This article refers to mainstream integenerational mobility studies and the processes we are studying at three different levels are the main point in the whole theoretical framework - starting from introduction.

Point 6: The authors discuss at the beginning of Section 3 on occupation and education systems of Nordic countries and Germany. I think the authors should describe in a more detailed manner Italian labor markets and education systems before comparing them to other countries. The beginning of this section is somewhat confusing because it does not mention table 1 it might refer to. Thus, it would be better first to describe the Italian system and then compare it to other countries. However, I think table 1 is a nice presentation of the labor market closure of the different occupations between the countries.  

I think authors should explain even better their target population. Who are the Italian graduates? You write that “we do not consider those graduates who only hold a BA degree”, thus my question is who are the graduates? You should clarify the population under the study. Also, if you only follow graduates from higher education your results cannot be generalized to the whole population. Thus, you should address that the study population is higher educated and their parents.

Response 6: we thank the reviewer for the comment. As already mentioned we decided to fill the lack related to higher level professionals, a specif sector of the Italian labour market we widely explain – mainly characterized by professional closure and lower level of internal competition. As the reviewer noticed, thank to the index and our work to replicate the level of closure for medical doctor we show a comparison between countries that is very easy to understand (from 0 to 6 where 0 means no closure and 6 the highest level of professional closure). Then the target population cannot be BA graduates but MA or complete degree of 6 years for doctors as the professional regulation imposes. Moreover, the regulation do not stop with the degrees in specific fields of study but other steps are requested as explain in the theoretical framework.

Point 7: At least in the logistic regressions, many control variables have been used. Authors should explain, what these control variables are in the data and methods section. Further, they should add the description of logistic regression and AME. I mean what are the models that you use. Please, clarify this.

Response 7: we added controls variables specification also in the data and methods sections – see line 337.

Minor comments

Point 8 : It would be more reader-friendly to separate the method section from the data section.

Can you clarify your figure 5 caption, is the y-axis always parental occupational standing? 

Response 8: This is specified in line 518

Point 9: There is a typo in the 5. title in the word gender.

Response 9: thank you. The typo has been corrected.

Reviewer 2 Report

This paper uses log-linear models applied to two Italian data sets to contrast intergenerational persistence across big classes, “meso-classes, and micro-classes, with a special focus on the service/professional class and on those occupations ("microclasses") within this EGP class that are regulated by the state through, for example, licensing or mandatory minimum credentials.

This paper holds a lot of promise. I especially appreciate the effort to operationalize indicators of occupational closure and embed them in more standard models of mobility, thus more directly tying these two closely related literatures together. That is, most of the early theoretical and empirical work on institutionalized forms of occupational closure, including licensing and regulation, focused on its effects on wages [e.g., Weeden 2002, Bol and Weeden 2013], while most of the empirical work on micro-class mobility [e.g., Grusky and Sorensen, Jonsson et al] has focused on the big/meso/micro mobility contrasts, but without attempting to explicitly measure the effects of occupational licensure and regulation on intergenerational micro-class mobility. The main contribution here is to bridge this gap. My overarching advice for the authors would be to downplay the big class vs. meso class vs. micro-class contrast, which I think takes up too much of the empirical attention (and, frankly, is getting to be a rather well-worn and insular set of debates) and instead highlight the regulated vs. unregulated contrast and any gender differences therein.

More specific comments:

1.              The meso classes, in particular, are undertheorized – a problem not unique to this paper, to be sure, but quite noticeable in it. The main issue is that the authors offer no theoretical justification for differentiating the higher service class (ala EGP) into professionals, managers, and entrepreneurs. The secondary issue, or perhaps the irony, is that the resulting “meso-class” scheme is nearly identical to the Featherman-Hauser big class scheme, which other studies have shown performs better than EGP (but worse than a micro-class scheme) in capturing class heterogeneity in all kinds of social and behavioral outcomes, including children’s attainment (see Weeden and Grusky 2005 for an analysis of this in the American context; Manza, Brooks, and Hout also made a similar argument about managers vs. professionals with respect to political behavior.) Differentiating self-employed and employed professionals, as in model 7a, further brings EGP into alignment with FH. We are left, then, with a “meso-class” scheme that is basically a relabeled – and unacknowledged -- version of Featherman-Hauser. All this said, as I mentioned above, I think the size of classes debate and analysis is the least interesting part of the paper, and should be minimized in its final version.

2.              Modeling

a.    The discussion of the log-linear analyses (what was done, and why) and the relationship of the models to the hypotheses was quite hard to follow, even for someone who is reasonably well versed in the mobility literature and trained in log-linear models. Just as one example, Hypothesis 1 is explicitly about downward mobility, which (a) explicitly assumes a hierarchical ordering of classes that is, firstly, at odds with Goldthorpe’s insistence that his scheme is not hierarchical, and secondly, not explicitly modeled in the current models, which treat all forms of mobility as equivalent. And, of course, any claim about why professional children want to become professionals (either in general or in microclasses) – such as the reference in the paper to "incentives" or to cognitive and cultural capital – cannot be tested with these data.

b.       There appear to be some typos in Table 2, assuming the “MX.x” indicators in parentheses in the “significance” column refer to the contrasting model. Specifically, Model 1b’s contrast is, according to the table, Model 1b. Model 3b’s contrast is listed as Model 6a, but I assume it was supposed to reference M3a.

c.     Some of the modeling decisions and choices of baseline models against which to contrast less parsimonious specifications could benefit from more in-depth explanation and justification, especially those in which self-employed professionals are contrasted with employed professionals at the micro-class level (models 7a and 7b). Weeden and Grusky wrestled with the treatment of the self-employed in big-class and micro-class schemes in the 2005 AJS paper referenced above; IIRC, their solution was to fit a self-employment shift effect on top of the micro-class effect. It would be useful for you to discuss how your modeling decisions differ from and improve on their strategy.  

d.       Related to this, I didn’t understand why the models with the interaction of regulation and self-employment (6a/6b) are presented before those with just self-employment. It could be that I’m not understanding the class level at which the self-employed effect is specified across the two models. Similarly, it’s not clear why model 3 is presented before model 4

e.   The authors have an occupation-level index of regulation, but – as far as I could tell – in the log-linear models collapse all index values into a binary indicator of whether a profession is regulated or not. This may be a justifiable simplification of the data, but at least on the surface it seems like it is an unnecessary loss of information that, in theory, should be associated with mobility chances. The figures show this association descriptively, but what is the argument against embedding the index values into the nested model contrast (e.g., using scaled association models)?

f.     Although it certainly has its critics, BIC is fairly standard in mobility research as a tool for model selection. The authors rely on the index of dissimilarity instead, but don’t explain why they ignore BIC. This becomes relevant insofar as BIC, which strongly favors parsimony, is likely to identify a very different preferred model than delta or L2 contrasts. 

g.     The switch to a logistic regression to model persistence in the higher service needs to be better justified or discarded. Any log-linear model can be rewritten as a logit model, and hence the information about the likelihood of persistence in the higher service class is embedded in the coefficients of the preferred log-linear model from Table 2. The “value added” of the logistic regression is thus that the immobility effects are adjusted for demographic covariates, but it’s not clear from the front end of the paper why this is important or necessary. My recommendation is to drop this analysis, and instead present the coefficients from the preferred log-linear model instead (normalized or from effect-coded parameterizations, so that the coefficients are not represented as deviations from an arbitrary baseline cell). This would also help to clean up a paper that already has a lot going on, making it easier for the reader to follow.

3.              The authors allude to the problem of differential selection into the sample, but I think downplay the extent to which selection may affect the results and limit the paper’s contribution to understanding mobility/immobility across different levels of aggregation and different class schemes. In terms of the extent of the selection, it was very clear (from the title of the paper) that the sample was limited to graduates, but I was surprised to learn midway through the paper that the analytic sample is also limited to college graduates with post-baccalaureate degrees (see lines 277-279). The analytic sample of the ILFI is limited to professionals (line 285). How is this not “sampling on the dependent variable”? How can one assess the relative propensity of children of fathers in the service class (or in professional “meso” or micro classes, or in the regulated professions, etc) to inherit their position or experience downward mobility if the sample is limited to professional children? (If I have misread the text and the sample is instead limited to professionals’ children, the nature of the selection problem is altered but it doesn’t go away.) I was not convinced, based on the text (and my memory of the mobility literature), that fitting marginal effects eliminates the problem of differential selection and its impact on the estimated immobility effects.

4.              Some of the more interesting and novel results in the paper pertain to the pattern of gender differences. However, I found the discussion and conclusion of these results somewhat thin, and in need of deeper engagement with the closure and gender inequality literatures.

a.     A consistent finding in the attainment and mobility literature, going back to the early Wisconsin studies in the late 1960s and early 1970s, is that the attainment of boys/young men is more strongly associated with social context effects of all sorts (family, neighborhood, geographic context, school attributes, etc) than the attainment of girls/young women. It would strengthen your paper if you discussed your results in the context of this pattern – where your results are consistent, where they are not, and what it might mean.

b.     In the discussion, the authors imply that more regulation could increase gender equality. Although the authors do not make the connection, this argument harkens back to Parkin, Murphy, and other early closure theorists, many of whom theorized that closure based on achieved statuses (educational credentials, licensure through examination) replaces and indeed delegitimizes closure based on ascribed statuses (that is, discrimination by gender, race, etc). This classic argument is in tension with more modern claims that that closure based on ascribed attributes can and does peacefully coexist with, or even support, closure based on achieved attributes. The paper would benefit from engaging this existing closure literature, since it’s relevant to the claims the authors are making about regulation and gender equality. Second, and related: it seems a bit simplistic to argue that more regulation will lead to more gender equality, both because of the two different theoretical claims about whether forms of exclusion compete, coexist, or complement/reinforce each other and because there’s no logical reason to assume a linear relationship between regulation and gender inequality. 

c.     Any weaker parental class or micro-class effect on young women’s attainment needs to be discussed with reference to gender segregation of occupations. In some sense, it is no surprise that an analysis of father-daughter mobility would show less micro-class inheritance than father-son mobility, simply because the many social processes that lead to gender segregation in occupations work “against” father-daughter occupational inheritance and “in favor of” father-son occupational inheritance. Gender segregation across big classes, as opposed to across occupations/micro-classes, is less extreme.

Typo in line 339: gender differences.

Author Response

Response to Reviewer 2 Comments

Before proceeding with my responses to the comments, I would like to thank the reviewers for the helpful suggestions and comments.

This paper uses log-linear models applied to two Italian data sets to contrast intergenerational persistence across big classes, “meso-classes, and micro-classes, with a special focus on the service/professional class and on those occupations ("microclasses") within this EGP class that are regulated by the state through, for example, licensing or mandatory minimum credentials.

This paper holds a lot of promise. I especially appreciate the effort to operationalize indicators of occupational closure and embed them in more standard models of mobility, thus more directly tying these two closely related literatures together. That is, most of the early theoretical and empirical work on institutionalized forms of occupational closure, including licensing and regulation, focused on its effects on wages [e.g., Weeden 2002, Bol and Weeden 2013], while most of the empirical work on micro-class mobility [e.g., Grusky and Sorensen, Jonsson et al] has focused on the big/meso/micro mobility contrasts, but without attempting to explicitly measure the effects of occupational licensure and regulation on intergenerational micro-class mobility. The main contribution here is to bridge this gap. My overarching advice for the authors would be to downplay the big class vs. meso class vs. micro-class contrast, which I think takes up too much of the empirical attention (and, frankly, is getting to be a rather well-worn and insular set of debates) and instead highlight the regulated vs. unregulated contrast and any gender differences therein.

More specific comments:

Point 1.              The meso classes, in particular, are undertheorized – a problem not unique to this paper, to be sure, but quite noticeable in it. The main issue is that the authors offer no theoretical justification for differentiating the higher service class (ala EGP) into professionals, managers, and entrepreneurs. The secondary issue, or perhaps the irony, is that the resulting “meso-class” scheme is nearly identical to the Featherman-Hauser big class scheme, which other studies have shown performs better than EGP (but worse than a micro-class scheme) in capturing class heterogeneity in all kinds of social and behavioral outcomes, including children’s attainment (see Weeden and Grusky 2005 for an analysis of this in the American context; Manza, Brooks, and Hout also made a similar argument about managers vs. professionals with respect to political behavior.) Differentiating self-employed and employed professionals, as in model 7a, further brings EGP into alignment with FH. We are left, then, with a “meso-class” scheme that is basically a relabeled – and unacknowledged -- version of Featherman-Hauser. All this said, as I mentioned above, I think the size of classes debate and analysis is the least interesting part of the paper, and should be minimized in its final version.

Response 1: we thank the reviewer for comment. We decided to employ the EGP class scheme (1992) because is widely employed in the social mobility literature and especially in the Italian case. However, only a few studies employed EGP meso-class differences within the upper class and as Ruggera and Barone (2017) and Ruggera (2016;2021) clearly pointed out that, especially for Italy, they are a basic starting point to show the great differences within the upper class, generally and too often conceptualized as a big-class only. We agree with the reviewer when it points out that Featherman – Hauser scheme perform better to analyse the American society, but we believe that employing it for the Italian case would make difficult for the reader to compare our results with the existing literature. As the reviewer noticed, however, the article and our operationalization do not miss to catch important dimensions allowing in depth specifications and/or explanations.

We furtherly define meso-classes since the introduction. Line 65-77- We also assert why we consider meso-classes and specifically the importance of the meso-class of professionals. As suggested by the reviewer, we also refer to Dalton R.J. and Klingemann H.D. (2007) and Lambert, P. and Griffiths, D. (2011) for a more comprehensive consideration of social and behevioural outcomes and related mechanisms.

When we operationalize social classes into details we want to relate to the existing literature on Italian intergenerational social mobility studies. As we point out (see also Schizzerotto 2002 or Barbagli and Schizzerotto), Erikson and Goldthorpe (1992) have distinguished the three meso-classes we employ, but only a few studies has used it (we first relate to the European societies on purpose. We strongly believe that EGP helps to relate to European societies and to improve social mobility literature in Italy. Additionally, as the reviewer pointed out, and that we thank for the comment, the professional closure opertationalization as well as the focus of the article is the heterogeneity of the upper class. Grusky and Weeden refer to class schemes that capture class heterogeneity in a more general way – considering immobility of other lower social classes, but missing the level of closure or other important variables (dimension) such as self-employment that in the EGP scheme or in the micro-class approach for professionals is not distinguished. We agree that EGP can and have to improve, which is why we specify meso, micro-classes and also by distinguishing the self-employment dimension that is a particular Italian labour market characteristic and particularity.

  1. Modeling

 there  is  abundant  evidence that

professionals  as  a  whole  differ  from  managers  and  entrepreneurs  in  terms of  value  orientations,

political  attitudes  and  lifestyles  (Dalton  and  Klingemann  2007).  Unsurprisingly,  professionals

associate more often with members of other professional groups (Lambert and Griffiths 2011)

there  is  abundant  evidence that

professionals  as  a  whole  differ  from  managers  and  entrepreneurs  in  terms of  value  orientations,

political  attitudes  and  lifestyles  (Dalton  and  Klingemann  2007).  Unsurprisingly,  professionals

associate more often with members of other professional groups (Lambert and Griffiths 2011

Point 2 a: The discussion of the log-linear analyses (what was done, and why) and the relationship of the models to the hypotheses was quite hard to follow, even for someone who is reasonably well versed in the mobility literature and trained in log-linear models. Just as one example, Hypothesis 1 is explicitly about downward mobility, which (a) explicitly assumes a hierarchical ordering of classes that is, firstly, at odds with Goldthorpe’s insistence that his scheme is not hierarchical, and secondly, not explicitly modeled in the current models, which treat all forms of mobility as equivalent. And, of course, any claim about why professional children want to become professionals (either in general or in microclasses) – such as the reference in the paper to "incentives" or to cognitive and cultural capital – cannot be tested with these data.

Response 2 a: We agree with the reviewer and we reformulated H1.Moreover, we added in the online appendix (before the references) the design matrixes to help the reader in the models understanding (in the first place, we provided these matrixes in a separate file). In this way, the modelling social immobility is clearer and the reader can follow hypotheses and modelling strategies into details. – Line 622 onwards.

Point 2b:  There appear to be some typos in Table 2, assuming the “MX.x” indicators in parentheses in the “significance” column refer to the contrasting model. Specifically, Model 1b’s contrast is, according to the table, Model 1b. Model 3b’s contrast is listed as Model 6a, but I assume it was supposed to reference M3a.

Response 2b: we changed Table 2 contrast for the homogeneous model indicated by the reviewer (for example 1b heterogeneous model is against 1a homogeneous model). Now we can clearly relate model a as homogeneous models and b as heterogeneous model with the corresponding contrasts.

Point 2 c:   Some of the modeling decisions and choices of baseline models against which to contrast less parsimonious specifications could benefit from more in-depth explanation and justification, especially those in which self-employed professionals are contrasted with employed professionals at the micro-class level (models 7a and 7b). Weeden and Grusky wrestled with the treatment of the self-employed in big-class and micro-class schemes in the 2005 AJS paper referenced above; IIRC, their solution was to fit a self-employment shift effect on top of the micro-class effect. It would be useful for you to discuss how your modeling decisions differ from and improve on their strategy. 

 Response 2 c: by adding the design matrixes the modelling strategy is now clarified. There was a misunderstanding in the online Appendix submission and we are sorry for the inconvenience. However, now each model design matrix is specified. We also furtherly clarified our models strategy to catch the self-employment at micro-level – model 7 a and 7b particularly as suggested by the reviewer. Lines 461 onwards. We agree with the reviewer, indeed, by adding this clarification on our modelling strategy helps the reader to better understand why we decided to catch each effect of employed and self-employed professionals group separately. In fact, we can interpret and compare each group effect since they have the same reference category. This is possible and helpful in our case because the focus is on the top of the occupational hierarchy; whereas Grusky and Weeden, have considered all diagonal cells, counting 82 micro-classes, which would make our strategy not suitable in that case – loosing the parsimony of log linear models. Moreover, going into details of these authors operationalization they did not compare self employed and employed professionals, which for the Italian case, as proven by our results analysis emerges as a very important specification. As pointed out in our theoretical framework, the institutional setting of Italy for licensed professionals is unique in Europe and worldwide. In the USA there are no more than 4 professions that can “resemble” licensed professionals in Italy. Certainly and unfortunately, advantages and professionals codes of conduct of Italian professionals are so strong and radiated that even after the law changes, “good behaviors ” these professionals just became highly recommended behaviors by their professional associations. This is even stronger for self employed professionals. This is why we strongly believe that this study can clearly underlie that belonging to different professions provide different levels of advantages based on the level of regulation as well as the opportunity to compete in the service sector as self-employed professionals.  

Point 2d :   Related to this, I didn’t understand why the models with the interaction of regulation and self-employment (6a/6b) are presented before those with just self-employment. It could be that I’m not understanding the class level at which the self-employed effect is specified across the two models. Similarly, it’s not clear why model 3 is presented before model 4

Response 2d: to the previous distinction models in Panel A and Panel B we added a specification in the table concerning Panel B models that specified self-employment and regulation dimensions combinations.

Point 2 e: The authors have an occupation-level index of regulation, but – as far as I could tell – in the log-linear models collapse all index values into a binary indicator of whether a profession is regulated or not. This may be a justifiable simplification of the data, but at least on the surface it seems like it is an unnecessary loss of information that, in theory, should be associated with mobility chances. The figures show this association descriptively, but what is the argument against embedding the index values into the nested model contrast (e.g., using scaled association models)?

Response 2e: we are really grateful to the reviewer who appreciate our work in considering the professional closure index – especially in the case of medical doctors that we build for different European countries even if they are basically licensed everywhere. And, we are sorry again for the misunderstanding regarding the appendix with each model design matrix. We calculated each professional group level of immobility separately; not only with SPL data but also with ILFI data on male population (see Appendix). All ILFI analysis and further specification of the analysis with SPL data are now available in the appendix. Analysis in SPL are confirmed with data that are representative of the entire population. From line 648 onwards. We agree that our figure has a genuine descriptive purpose being a bivariate correlation. We also believe that other kind of models and analyses would be useful to understand the effect of professional closure through the closure index. But, more importantly, as the reviewer already said, longitudinal surveys with detailed ISCO codes for both parents and children would be needed (eg. by using a counterfactual design). Another solution would be to insert the index at a different level, comparing Italy with other European countries by using multilevel models. But, in this case the existing surveys even if pooled together do not allow employing micro-classes for Italy. However, since this data offer a unique opportunity to study the upper class intergenerational mobility going into details of micro-classes for the Italian case, to compare SPL with ILFI data results we opted for these log-linear models with the same design matrixes. This is our choice to offer consistent and more understandable results to the reader.  

    Point 2 f : Although it certainly has its critics, BIC is fairly standard in mobility research as a tool for model selection. The authors rely on the index of dissimilarity instead, but don’t explain why they ignore BIC. This becomes relevant insofar as BIC, which strongly favors parsimony, is likely to identify a very different preferred model than delta or L2 contrasts. 

Response 2f: in this case we explained that we prefer the dissimilarity index over BIC because of te large size of the sample we analyse, furtherly offering a contrast between models based on L2 differences.

Point 2g:     The switch to a logistic regression to model persistence in the higher service needs to be better justified or discarded. Any log-linear model can be rewritten as a logit model, and hence the information about the likelihood of persistence in the higher service class is embedded in the coefficients of the preferred log-linear model from Table 2. The “value added” of the logistic regression is thus that the immobility effects are adjusted for demographic covariates, but it’s not clear from the front end of the paper why this is important or necessary. My recommendation is to drop this analysis, and instead present the coefficients from the preferred log-linear model instead (normalized or from effect-coded parameterizations, so that the coefficients are not represented as deviations from an arbitrary baseline cell). This would also help to clean up a paper that already has a lot going on, making it easier for the reader to follow.

Response 2g: We agree with the reviewer according to the suggestion of a further specification regarding logistic regressions (why we employ logistic regression and the related interpretation). lines 342 – 348. We understand that the review prefer our micro-class analysis over the meso-classes that we further specify in terms of self-employment specifically referring to the meso-class of professionals (and we are thankful for that), but the results in both the log-linear models and logistic regressions bring us to that consideration. Thus, we cannot discard the meso-class differentiation. As we mentioned before this is a characteristic of the Italian labour market and service sector regulation.   

Point 3:      The authors allude to the problem of differential selection into the sample, but I think downplay the extent to which selection may affect the results and limit the paper’s contribution to understanding mobility/immobility across different levels of aggregation and different class schemes. In terms of the extent of the selection, it was very clear (from the title of the paper) that the sample was limited to graduates, but I was surprised to learn midway through the paper that the analytic sample is also limited to college graduates with post-baccalaureate degrees (see lines 277-279). The analytic sample of the ILFI is limited to professionals (line 285). How is this not “sampling on the dependent variable”? How can one assess the relative propensity of children of fathers in the service class (or in professional “meso” or micro classes, or in the regulated professions, etc) to inherit their position or experience downward mobility if the sample is limited to professional children? (If I have misread the text and the sample is instead limited to professionals’ children, the nature of the selection problem is altered but it doesn’t go away.) I was not convinced, based on the text (and my memory of the mobility literature), that fitting marginal effects eliminates the problem of differential selection and its impact on the estimated immobility effects.

Response 3: we do not select on the dependent variable. As we specify in the text by focusing on specific professional group regulation we must consider those graduates with the entire long degree as 6 years for doctors or MA degrees for the other professionals. We analyze all those graduates with MA degrees because by adding BA degrees would be affecting the comparison we make. To be consistent we need to compare MA graduates only since BA graduates are not allowed entering into Italian regulated professions. However, we report in the appendix the distribution of the main variables with the mentioned specifications, in order to be consistent on this point as well.   

Point 4a:            Some of the more interesting and novel results in the paper pertain to the pattern of gender differences. However, I found the discussion and conclusion of these results somewhat thin, and in need of deeper engagement with the closure and gender inequality literatures.     A consistent finding in the attainment and mobility literature, going back to the early Wisconsin studies in the late 1960s and early 1970s, is that the attainment of boys/young men is more strongly associated with social context effects of all sorts (family, neighborhood, geographic context, school attributes, etc) than the attainment of girls/young women. It would strengthen your paper if you discussed your results in the context of this pattern – where your results are consistent, where they are not, and what it might mean.  In the discussion, the authors imply that more regulation could increase gender equality. Although the authors do not make the connection, this argument harkens back to Parkin, Murphy, and other early closure theorists, many of whom theorized that closure based on achieved statuses (educational credentials, licensure through examination) replaces and indeed delegitimizes closure based on ascribed statuses (that is, discrimination by gender, race, etc). This classic argument is in tension with more modern claims that that closure based on ascribed attributes can and does peacefully coexist with, or even support, closure based on achieved attributes. The paper would benefit from engaging this existing closure literature, since it’s relevant to the claims the authors are making about regulation and gender equality. Second, and related: it seems a bit simplistic to argue that more regulation will lead to more gender equality, both because of the two different theoretical claims about whether forms of exclusion compete, coexist, or complement/reinforce each other and because there’s no logical reason to assume a linear relationship between regulation and gender inequality.  Any weaker parental class or micro-class effect on young women’s attainment needs to be discussed with reference to gender segregation of occupations. In some sense, it is no surprise that an analysis of father-daughter mobility would show less micro-class inheritance than father-son mobility, simply because the many social processes that lead to gender segregation in occupations work “against” father-daughter occupational inheritance and “in favor of” father-son occupational inheritance. Gender segregation across big classes, as opposed to across occupations/micro-classes, is less extreme.

Point 4: we thanks the reviewer for the suggestion. In addition to our literature review , we specified other studies results on gender differences (which some of them are dated 2021; 2016) and the link with professional closure 411-429 (2012) specifically refereeing to the Italian case. This article wants to fill the lack in studies regarding professional/licensed professional thus we looked for the proper data which allows going into details of micro-classes at least within the heterogeneous Service class. This line is highly suggested by Barone and colleagues not only for the Italian case. Moreover we refer, as the reviewer suggested, to mother and daughter mobility in the conclusion when we suggested for further research on this topic to consider mother social class position. line 643

Further, since closure has been widely treated, and the reviewer suggested another specification of closure we decided to insert a further discussion of the social and professional closure in the appendix. In this way we allow the text not to become too large and difficult to follow, and at the same time, for those who want to know more on the concept of closure can find a further discussion that the reader can exploit for example for further works and/or knowledge – Line  674-689

Typo in line 339: gender differences.     Thanks again. The typo has been corrected.

Round 2

Reviewer 2 Report

None of the leftover issues affect the substance of the results or interpretation, nor do they warrant holding up publication of what is a solid paper and contribution to the mobility literature.

Minor point: heading of section "Why do sons and daughters of professionals feel inclined to become profes- 88
sionals?" should in my opinion be changed to "Why might sons and daughters [or just "children"] of professionals become professionals." The data offer no direct measures of what the adult children feel or aspire to do, just what they actually do. The wording of the question in the heading (as written now) assumes facts not in evidence.